# Knockout of liver fluke granulin, *Ov-grn-1*, impedes malignant transformation during chronic infection with *Opisthorchis viverrini*

**Sujittra Chaiyadet**[1,2☯], **Sirikachorn Tangkawattana**[3☯], **Michael J. Smout**[4☯], **Wannaporn Ittiprasert**[5☯], **Victoria H. Mann**[5], **Raksawan Deenonpoe**[6], **Patpicha Arunsan**[1,5], **Alex Loukas**[4]*, **Paul J. Brindley**[5]*, **Thewarach Laha**[1]*

1 Department of Parasitology, Faculty of Medicine, Khon Kaen University, Khon Kaen, Thailand, 2 Tropical Medicine Graduate Program, Academic Affairs, Faculty of Medicine, Khon Kaen University, Khon Kaen, Thailand, 3 Faculty of Veterinary Medicine, Khon Kaen University, Khon Kaen, Thailand, and WHO Collaborating Center for Research and Control of Opisthorchiasis, Tropical Disease Research Center, Khon Kaen University, Khon Kaen, Thailand, 4 Centre for Molecular Therapeutics, Australian Institute of Tropical Health and Medicine, James Cook University, Cairns, Queensland, Australia, 5 Department of Microbiology, Immunology and Tropical Medicine, and Research Center for Neglected Diseases of Poverty, School of Medicine & Health Sciences, George Washington University, Washington, District of Columbia, United States of America, 6 Department of Pathology, Faculty of Medicine, Khon Kaen University, Khon Kaen, Thailand

☯ These authors contributed equally to this work.
* alex.loukas@jcu.edu.au (AL); pbrindley@gwu.edu (PJB); thewa_la@kku.ac.th (TL)

**Data Availability Statement:** The sequence reads are available at GenBank BioProject PRJNA385864, Sequence Read Archive study SRP110673,

## Abstract

Infection with the food-borne liver fluke *Opisthorchis viverrini* is the principal risk factor for cholangiocarcinoma (CCA) in the Mekong Basin countries of Thailand, Lao PDR, Vietnam, Myanmar and Cambodia. Using a novel model of CCA, involving infection with gene-edited liver flukes in the hamster during concurrent exposure to dietary nitrosamine, we explored the role of the fluke granulin-like growth factor *Ov*-GRN-1 in malignancy. We derived RNA-guided gene knockout flukes (*ΔOv-grn-1*) using CRISPR/Cas9/gRNA materials delivered by electroporation. Genome sequencing confirmed programmed Cas9-catalyzed mutations of the targeted genes, which was accompanied by rapid depletion of transcripts and the proteins they encode. Gene-edited parasites colonized the biliary tract of hamsters and developed into adult flukes. However, less hepatobiliary tract disease manifested during chronic infection with *ΔOv-grn-1* worms in comparison to hamsters infected with control gene-edited and mock-edited parasites. Specifically, immuno- and colorimetric-histochemical analysis of livers revealed markedly less periductal fibrosis surrounding the flukes and less fibrosis globally within the hepatobiliary tract during infection with *ΔOv-grn-1* genotype worms, minimal biliary epithelial cell proliferation, and significantly fewer mutations of *TP53* in biliary epithelial cells. Moreover, fewer hamsters developed high-grade CCA compared to controls. The clinically relevant, pathophysiological phenotype of the hepatobiliary tract confirmed a role for this secreted growth factor in malignancy and morbidity during opisthorchiasis.

sequence runs SRR15906234-15906251, accessions SRX12196673-SRX12196690.

**Funding:** This study was supported by the Research Program, Research and Graduate Studies, Khon Kaen University to SC, ST and TL, the National Cancer Institute, National Institutes of Health (NIH) USA award R01CA164719 (AL, TL, PJB,) and the Australian National Health and Medical Research Council, NHMRC award APP1085309 to AL and TL, and senior principal research fellowship APP1117504 to AL. This research was funded in part, by the Wellcome Trust, grant number 107475/Z/15/Z to PJB. The funders had no role in study design, data collection and analysis, decision to publish, or preparation of the manuscript.

**Competing interests:** The authors have declared that no competing interests exist.

## Author summary

Infection with the human liver flukes, *Opisthorchis viverrini*, *O. felineus* and *Clonorchis sinensis* remains a public health concern in regions where these parasites are endemic. *O. viverrini* is endemic in the Mekong River drainage countries including Thailand and the Lao People's Democratic Republic. Infection follows the consumption of undercooked freshwater fish harboring the parasite. Liver fluke infection, opisthorchiasis, is associated with diseases of the liver and bile ducts including cancer of the biliary tract, cholangiocarcinoma, a cancer with a poor prognosis. This report characterizes, for the first time, experimental infection with gene-edited *O. viverrini* liver flukes during concurrent exposure to a dietary nitrosamine in a rodent model of liver fluke infection-associated cancer. Cancer development was slowed in hamsters infected with the parasite following CRISPR-based knock-out mutation and loss of a parasite gene known to stimulate growth of cells lining the bile ducts. These findings definitely link a parasite factor to cancer etiology, and present a new laboratory model to investigate risk factors for infection-associated cholangiocarcinoma and to assess efficacy of anti-infection/anti-cancer vaccines.

## Introduction

Liver fluke infection caused by species of *Opisthorchis* and *Clonorchis* remains a major public health problem in East Asia and Eastern Europe. Infection with *Opisthorchis viverrini* is endemic in Thailand and Laos, where ~10 million people are infected with the parasite. Opisthorchiasis is associated with hepatobiliary diseases including cholangiocarcinoma (CCA), bile duct cancer [1,2]. Northeast Thailand reports the world's highest incidence of CCA, > 80 per 100,000 in some provinces. Indeed, the International Agency for Research on Cancer of the World Health Organization classifies infection with *O. viverrini* as a Group 1 carcinogen, i.e. definitely carcinogenic in humans [1,3,4].

Which features or consequences of parasitism by the liver fluke definitely initiate malignant transformation to CCA have yet to be ascertained notwithstanding that opisthorchiasis is the principal risk factor for CCA in regions where this neglected tropical disease remains endemic [1, 4–6]. Some factors can be expected to more important than others and the impact of these factors should be quantifiable. Worm burden plays a role, based on rodent models of liver fluke infection-associated CCA [7], as does concurrent exposure to nitrosamines in the fermented foods [8,9] that are culturally important dietary staples in countries of the Lower Mekong River basin [5]. Moreover, dose-dependent, synergistic effects of the liver fluke and nitroso compounds have been documented [7,10]. To survive within the host, parasitic helminths actively release excretory/secretory (ES) proteins and other mediators with diverse effects and roles at the host-parasite interface [11,12]. This interaction is considered to manipulate host cellular homeostasis and, moreover, to underpin malignant transformation during chronic opisthorchiasis, but the molecular mechanisms by which these processes operate remain inadequately understood [13]. Focusing on the contribution of liver fluke ES to carcinogenesis, we targeted the *O. viverrini* granulin-like growth factor, *Ov*-GRN-1, a prominent component of the ES complement that we had determined previously induces phenotypic hallmarks of cancer [14,15]. *Ov*-GRN-1 and other ES components including extracellular vesicles enter cholangiocytes, the epithelial cells that line the biliary tract, and appear to drive cellular signaling that promotes carcinogenesis, including cellular proliferation and migration, angiogenesis and wound healing [16]. We have confirmed the role of *Ov*-GRN-1 in driving proliferation of bile duct epithelial cells (cholangiocytes) by genetic manipulation of its expression in

the liver fluke both by RNA interference and RNA-guided gene knockout [15,17,18]. Moreover, we have shown that infection of hamsters with the gene edited, infectious stage of the live fluke was feasible and that proliferation of biliary epithelia is markedly suppressed during infection with the *ΔOv-grn-1* (*Ov-grn-1* knockout) flukes [18].

There is an established and instructive model of the pathogenesis of CCA in experimentally infected hamsters, *Mesocricetus auratus* that is considered to replicate the epidemiology and pathogenesis of chronic human opisthorchiasis [1]. In this model, malignancy manifests within a few months following infection with metacercariae of the parasite and concurrent exposure to otherwise sub-carcinogenic levels of dietary nitrosamine [7,19,20]. In the hamster, chronic opisthorchiasis provokes periductal fibrosis, which, coincident with exposure to the nitric oxide carcinogen facilitates cholangiocyte proliferation, epithelial hyperplasia, DNA damage, and allied biliary tract lesions [21], which can culminate in CCA [22]. Using this model, here we investigated the outcome of infection of hamsters with gene-edited *O. viverrini* liver flukes in relation to fluke-induced periductal fibrosis and malignant transformation. For this investigation, hamsters were infected with juvenile flukes that had been genetically modified using CRISPR before infection. Specifically, following up on our earlier study [18] which focused on knockout of the *Ov-grn-1* gene, here we also included, as comparators, juvenile flukes gene-edited for a second virulence factor, tetraspanin (*Ov-tsp-2*) [23], which also networks at the host-parasite interface [24,25] and, as controls, flukes subjected to CRISPR transfection with an irrelevant (non-targeting) guide RNA [26].

Immuno- and colorimetric-histochemical analysis of thin sections of hamster liver revealed markedly less fibrosis during infection with *ΔOv-grn-1* worms, reduced proliferation of cholangiocytes, substantially less expression of mutant forms of the p53 tumor suppressor protein and, overall, diminished malignancy of the liver. The clinically relevant, pathophysiological phenotype confirmed a role for *Ov*-GRN-1 in morbidity and malignancy during opisthorchiasis. In addition, these findings underscored the utility and tractability of CRISPR-based genome editing for addressing gene function, essentiality, and pathogenesis in parasitic helminths generally.

## Results

### Hamster model of malignant transformation during infection with gene-edited *Opisthorchis viverrini*

To investigate the effect of programmed gene knockout (KO) in *O. viverrini*, two experiments were undertaken in which hamsters were infected with newly excysted juveniles (NEJs) of *O. viverrini* that had been subjected to programmed KO. The CRISPR/Cas materials were delivered by electroporation of plasmids encoding guide RNAs specific for *Ov-grn-1*, *Ov-tsp-2* and an irrelevant (control) guide RNA. All groups received plasmids encoding the Cas9 nuclease from *Streptococcus pyogenes*. Fig 1A illustrates the experimental approach and timelines, the findings from which we present below. Initially, juvenile flukes were subjected to transfection after which reduction in transcription of the targeted genes, *Ov-grn-1* and *Ov-tsp-2*, was verified. Subsequently, following successful KO of transcription *in vitro*, additional juvenile *O. viverrini* were transfected with the CRISPR plasmids before infection of hamsters. The goal of Experiment 1 was to assess impact of KO on the worm burden. At necropsy at 14 weeks after infection, the gross appearance of the liver was examined, the worms were recovered and counted, and transcriptional changes were investigated in the worms. With Experiment 2, the primary goal was establishment and assessment of disease burden including malignant transformation of the liver during infection. Given that livers from the hamsters in Experiment 2 were fixed at necropsy, worm burdens could not be established directly because recovery of

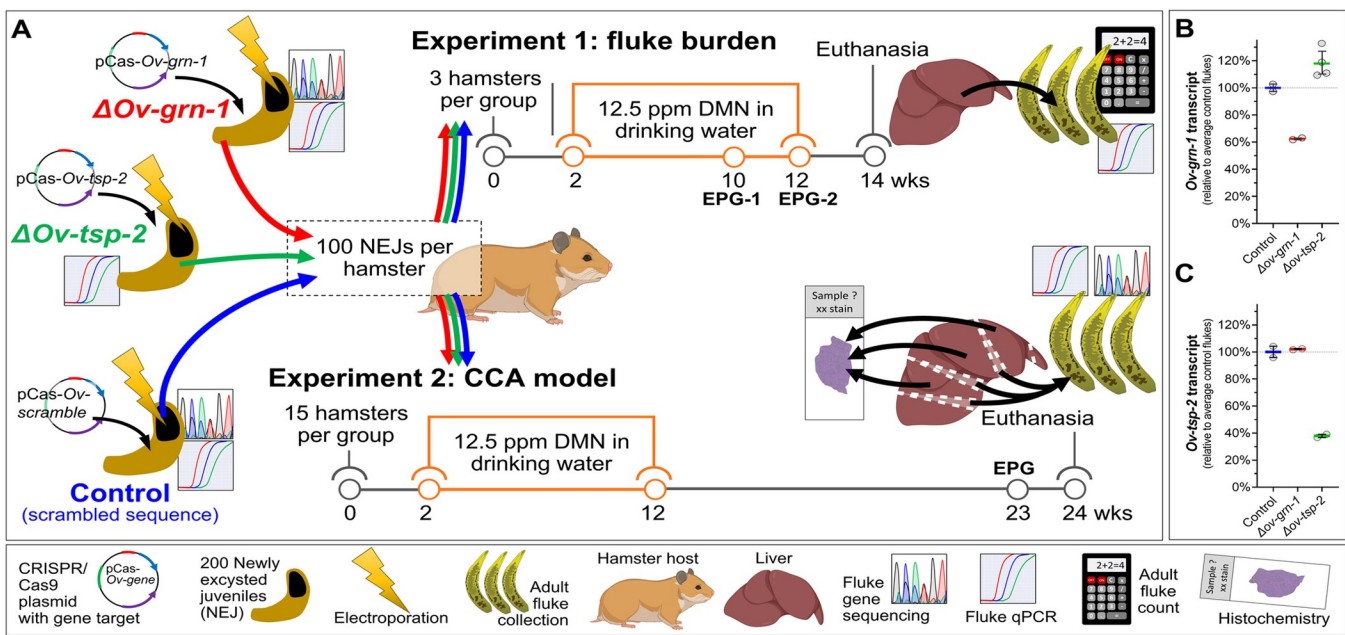

**Fig 1. Schematic overview of the experimental design.** Hamsters were infected with juvenile *Opisthorchis viverrini* worms that had been subjected to gene knockout. Two experiments were conducted. (**A**). The goal of experiment 1 was to assess the impact of CRISPR/Cas9 editing on fluke survival. The goal of experiment 2 was to assess the influence of gene knockout on pathogenesis and malignant transformation. Groups of hamsters were infected with flukes transfected with CRISPR/Cas9 plasmids targeting either *Ov-grn-1* (red, *ΔOv-grn-1*), *Ov-tsp-2* (green, *ΔOv-tsp-2*), or irrelevant guide RNA (blue, control) and exposed to dimethyl nitrosamine (DMN) in the drinking water. At timepoints indicated for each experiment, fecal egg numbers were measured as eggs per gram of feces (EPG). Experiment 1: From the livers of euthanized hamsters, all the flukes were recovered and transcript levels of the targeted genes assessed. Experiment 2: Liver lobes were sectioned and stained for histochemical analysis. Flukes incidentally released from bile ducts were collected and assessed for gene knockout. Before infection, transcript levels of *Ov-grn-1* (**B**) and *Ov-tsp-2* (**C**) were assessed in juvenile flukes. Transcript levels established by qPCR were plotted relative to average control transcript levels from 2–4 biological replicates; average shown with colored bar and with 95% confidence interval bars. Population statistics were generated from resampling 1000 times by replacement bootstrap analysis of untransformed delta-delta Ct values (derived from S1 Fig). (Elements of the figure include icons from BioRender.com with copyright and licensing permission).

the flukes is a process that damaged the liver and biliary tract and, accordingly, was incompatible with histological assessment of infection-associated hepatic disease. Consequently, only a small number flukes were available from Experiment 2 but those that were available were collected incidentally during preparation of the liver lobes for fixation. Nonetheless, this sample of the flukes from each of the three groups was sufficient to assess the performance and level of programmed KO although total worm burden was unavailable. Nonetheless, the findings from Experiment 2 provided the first description of gene-edited flukes in chronically-infected hamsters, i.e., with duration of infection beyond eight weeks and, additionally, the first time programmed mutation of the liver fluke genome has been combined with exposure to dietary nitrosamine in the hamster-liver fluke model of human CCA.

Changes in transcription of the targeted genes induced by programmed mutations were monitored by RT-qPCR in newly excysted juveniles (NEJs) (S1 Fig). Control (irrelevant gRNA) group flukes were used to normalize expression level from 100% (no change) to 0% (complete ablation of expression) (Fig 1B and 1C). For the *ΔOv-grn-1* group, *Ov-tsp-2* was used as an off-target gene control, and *Ov-grn-1* was used as an off-target control for the *ΔOv-tsp-2* group. Relative to the control group, *Ov-grn-1* transcript levels were substantially reduced in the *ΔOv-grn-1* flukes (bootstrapped average, 95% confidence interval [CI]: 62.4%, 61.8–63.1%) whereas the transcription of *Ov-tsp-2* increased marginally (< 20% change) in the *ΔOv-grn-1* flukes (117.8%, 95% CI: 110.2–127.0%) (Fig 1B). Transcription of *Ov-tsp-2* was substantially reduced in the *ΔOv-tsp-2* flukes (37.9%, 95% CI: 36.7–39.1%). Levels of *Ov-grn-1*

transcripts (102.1%, 95% CI: 101.8–102.5%) were unchanged (Fig 1C). This outcome revealed gene-specific, on-target knockout at the *Ov-grn-1* and *Ov-tsp-2* loci. Gene expression analyses were restricted to the two targeted genes and we cannot conclude that off-target mutations did not occur.

## Differential outcomes for adult flukes following knockout of *Ov-grn-1* and *Ov-tsp-2*

In Experiment 1, feces were sampled at both weeks 10 and 12 after infection (Fig 1A). Significant differences in fecal egg counts (EPG) were not apparent among the three groups (Fig 2A). Hamsters were euthanized and necropsied at week 14 to investigate the numbers of adult *O. viverrini*. There were 60.0±3.46 (mean ± SEM), 36.7±3.48, and 21.3±2.96 worms in the control, *ΔOv-grn-1*, and *ΔOv-tsp-2* groups, respectively, reflecting reductions of 38.9% for *Ov-grn-1* and 64.5% for *Ov-tsp-2* (Fig 2B). A trend was apparent toward increased egg burdens in hamsters with higher worm burdens but the correlation not statistically significant (S2 Fig).

Of the worms recovered in Experiment 1, gene transcript levels relative to the control flukes were assessed for 10–13 flukes from each of the three hamsters in each group. Transcript levels (S3 Fig) of both genes expressed by the control parasites clustered around 100% of wild-type fluke expression levels with bootstrap averages of 114.1% (95% CI: 104.2–126.5%) and 103.2% (95% CI: 98.4–107.5%) (Fig 2C and 2D). Transcript levels for *Ov-grn-1* ranged broadly in the *ΔOv-grn-1* flukes from no change (~100%) to complete ablation (~0%) and, overall, were substantially reduced at 11.8% (95% CI: 4.5–28.8%) of levels seen in wild-type flukes (Fig 2C). This phenotypic range, no change to ablation of transcripts in *Ov-grn-1*, was similar to our previous findings [18]. In contrast to the juvenile flukes, where there was substantial knockdown of *Ov-tsp-2* (Fig 1C), transcript levels were not markedly reduced in adult *ΔOv-tsp-2* flukes (82.5%, 95% CI: 73.6–91.5%) compared to the controls (103.2%, 95% CI: 98.4–103.2%) (Fig 2D). Indeed, most adult flukes in the *ΔOv-tsp-2* group were unchanged with only two individual flukes exhibiting > 50% reduction in *Ov-tsp-2* expression. We posit that this marked difference, in comparison to transcript levels for *Ov-tsp-2* in juvenile flukes, i.e. reduced by 62.1%, in addition to the 65% reduction of worms recovered at necropsy, *Ov-tsp-2* KO led to a lethal phenotype and that flukes of this genotype failed to survive *in vivo*. This contrasted with *Ov-grn-1* KO where the majority of flukes of the *ΔOv-grn-1* genotype survived even though transcription of *Ov-grn-1* was not detected in four of 36 flukes, and 10 of 36 exhibited *Ov-grn-1* transcript levels < 5% of levels expressed by the control worms.

## Parasitological impact of gene knockout

A primary goal for Experiment 2 was the quantitative assessment by histopathological and histochemical analysis of hepatobiliary disease at 24 weeks of infection. Accordingly, determining the number of surviving liver flukes was not feasible because the liver and its intrahepatic biliary tract, occupied by the parasites, was necessarily fixed in formalin at necropsy for downstream histological processing and analysis of the fixed liver sections. Nonetheless, we collected a sample of the resident *O. viverrini* flukes before formalin fixation of the liver for molecular screening to assess the efficacy of programmed gene knockout. As an accepted correlate of the number of worms parasitizing each hamster [27], fecal egg counts (as eggs per gram of feces, EPG) were determined at the time of necropsy. EPG values ranged broadly among the groups (Fig 3A), with feces of the control fluke infected hamsters displaying the highest EPG values, median of 11,062 EPG (95% CI: 4,814–26,536), the *ΔOv-grn-1* group, 5,267 EPG (95% CI: 2,695–8535), and the *ΔOv-tsp-2 group*, 4,530 EPG (95% CI: 518–10,227), i.e. in the same rank order in numbers of worms recovered from these groups in Experiment 1

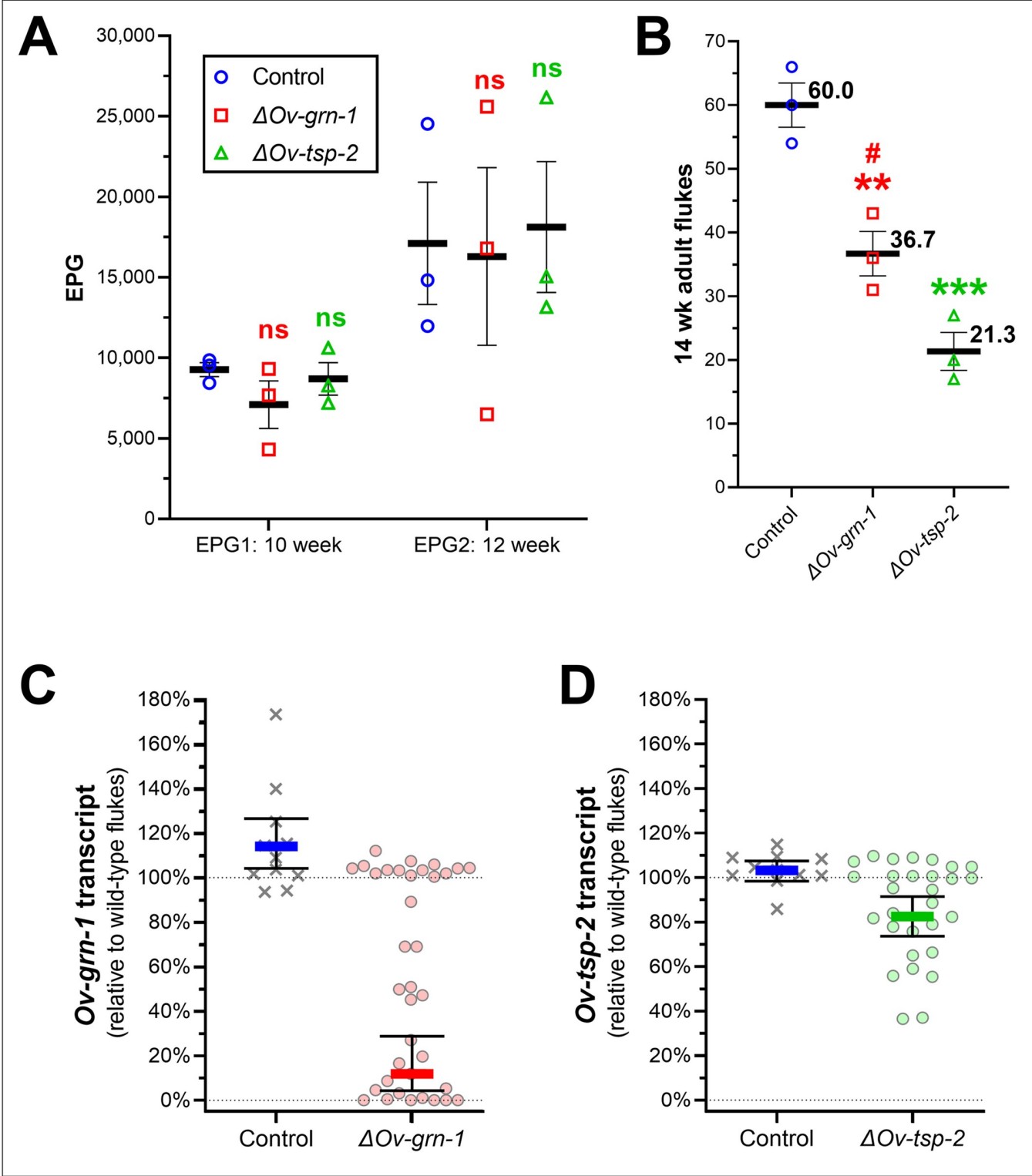

**Fig 2. Liver fluke burden and levels of gene transcription.** Fecundity, worm numbers, and gene expression levels were determined at 10–14 weeks after infection of the hamsters with 100 gene edited juveniles, and from three hamsters per group (Experiment 1). Number of eggs per gram of feces (EPG) from each hamster at weeks 10 and 12 (**A**) and worm numbers at week 14 (**B**) showing mean (horizontal black line) and SEM bars. Each treatment group was compared to the control group in 2-way ANOVA with Holm-Sidak multiple comparison: ns = not significant; **, $P \leq 0.01$; ***, $P \leq 0.001$, and $\Delta Ov\text{-}grn\text{-}1$ against $\Delta Ov\text{-}tsp\text{-}2$: #, $P \leq 0.05$. Gene transcript levels of $Ov\text{-}grn\text{-}1$ (**C**) and $Ov\text{-}tsp\text{-}2$ (**D**) were determined by qPCR for 10 to 13 flukes sampled from each animal

(30–39 flukes total per group) and plotted with each datum point representing the transcript level of an individual fluke relative to wild-type flukes. Resampling with replacement bootstrap analysis (B = 1000) of ddCT scores (S3 Fig) was used to generate population average, as denoted by the thick, colored line and 95% confidence interval bars.

(Fig 2B). All the groups showed substantial variation; the average EPG of *ΔOv-grn-1* hamsters was not significantly different to controls whereas the median EPG of the *ΔOv-tsp-2* group was significantly lower than the control group *(P ≤ 0.05)*.

Adult worms recovered from Experiment 2 hamster livers examined at 24 weeks after infection were evaluated for targeted gene transcripts by qPCR. *Ov-grn-1* transcript levels (S4 Fig) of adult *O. viverrini* from *ΔOv-grn-1* infected hamsters were substantially decreased down to a bootstrapped average of 10.6% relative to wild-type flukes (95% CI: 3.1–30.9%) compared to the control group with 91.0% (95% CI: 84.1–97.5%) (Fig 3B). In contrast, numbers of flukes in the *ΔOv-tsp-2* treatment group were similar, at 101.1% (95% CI: 91.1–111.3%) of wild-type flukes 98.5% (95% CI: 82.3–114.6%) of the controls (Fig 3C). With Experiments 1 and 2 showing the surviving *ΔOv-tsp-2* group worms expressing *Ov-tsp-2* at levels comparable to the control flukes, we posited that substantial *Ov-tsp-2* gene edits were lethal, and worms that survived to maturity likely had not undergone gene editing and/or few of the cells in the worms had been edited. Although we retained the *ΔOv-tsp-2* group for comparison of pathogenesis, the genotype of the flukes from this group was not investigated further.

## Synopsis of outcomes of CRISPR/Cas9 gene editing of the liver flukes

To characterize mutations from the programmed knockouts, a region of 173 bp flanking the programmed cleavage site in *Ov-grn-1* was scrutinized by analyzing aligned read-pairs from targeted amplicon NGS and analysis of the aligned read-pairs with CRISPResso2 [28].

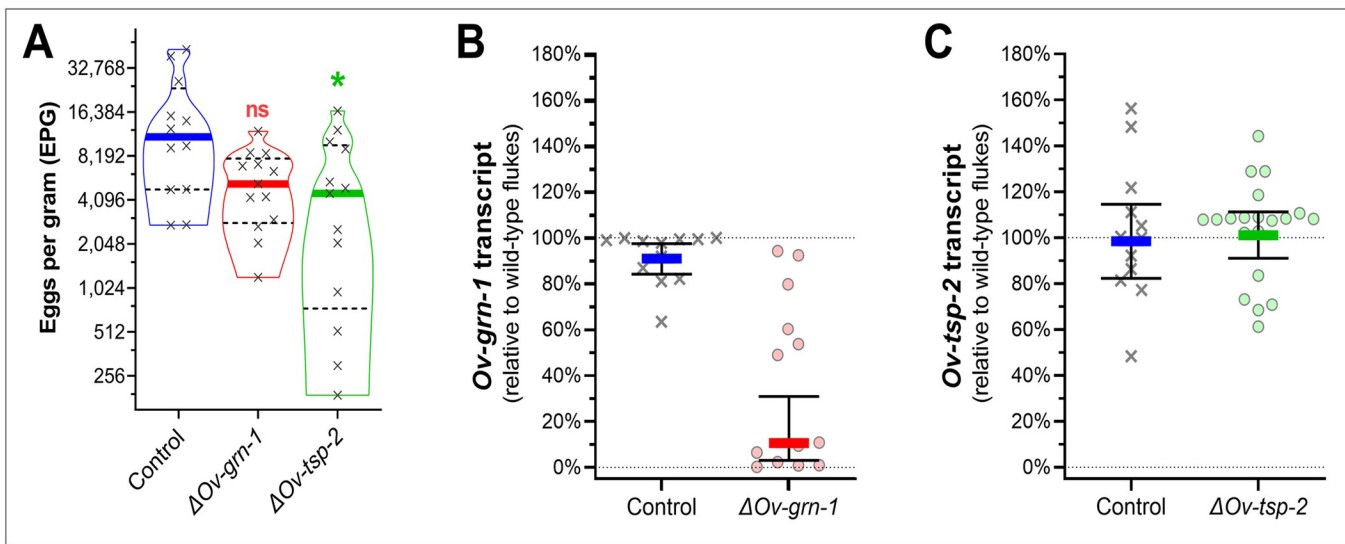

**Fig 3. Cholangiocarcinoma model, fecundity, gene transcript and mutation rates.** Eggs per gram of feces (EPG) were assessed at week 23 prior to euthanasia at week 24 (Experiment 2). Panel **A**, EPG values of the three groups of hamsters. Violin plot denotes each hamster's EPG with "x" symbols. Solid colored lines indicate the median values and dashed black lines indicate the quartiles. Kruskal Wallis with Dunn's multiple comparison correction was used to compare EPG levels against control group: *, P ≤ 0.05; ns, not significant. At necropsy, a sample of 12 to 20 flukes were collected from each group, transcript levels determined for *Ov-grn-1* (**B**) and *Ov-tsp-2* (**C**), and plotted with each datum point representing the transcript level of the individual fluke relative to wild-type flukes. Resampling with replacement bootstrap analysis (B = 1000) of ddCT scores (derived from S4 Fig) was used to generate population average–denoted by the thick, colored line and 95% confidence interval bars.

Substitution patterns as determined by the CRISPR-sub tool [29] in the KO groups were not significantly different from the cognate alleles in the controls (S5 Fig). Also, we scanned insertions and deletions (indels) and, in turn, the potential impact of indels on the open reading frame. Fig 4A and S1 Table present the indel percentages of juvenile and adult flukes. The *ΔOv-grn-1* pooled NEJs showed 3.26% indel levels (2,723 of 80,571 aligned read-pairs), significantly more than the control group (18 of 51,402 aligned read-pairs, 0.035%) ($P \leq 0.05$). The juvenile and adult flukes in the control group showed similar indel % levels, with 0.045% in the adults (41 of 91,783 aligned read-pairs). Individual *ΔOv-grn-1* adult *O. viverrini* flukes displayed a broad range of editing efficiency in terms of indel profiles. These ranged from an apparent absence of programmed mutation (no indels) to near complete KO (91% indels), with a median of 3.1% indels ($_M$*ΔOv-grn-1*), which was significantly higher than in the control group flukes ($P \leq 0.01$). As noted for levels of transcription, however, there were apparently distinct groupings consisting of six flukes of low level mutation status (termed $_L$*ΔOv-grn-1*) and six highly mutated flukes ($_H$*ΔOv-grn-1*) observed. From a total of 711 megabase pairs sequenced with 1.7 million aligned read pairs, programmed deletions (0.5 million) were overwhelmingly more common than insertions, with only seven insertions identified (S1 Table). There was an inverse correlation between efficiency of KO of *Ov-grn-1* (indel percentage), with a two-tailed non-parametric Spearman correlation co-efficient $r_s$ = -0.74 (Fig 4B; $P \leq 0.01$). Minimal expression only of *Ov-grn-1* was detected in the highly mutated, $_H$*Ov-grn-1* flukes, < 11% level of transcription of *Ov-grn-1* of control liver flukes. However, the highly edited genotype/highly reduced transcription phenotype contrasted with the wide range of transcription in the flukes with low or moderate levels of editing, $_L$*ΔOv-grn-1* and $_M$*ΔOv-grn-1*, with a wide range of transcription from 6% to 94% of the levels of the control group worms.

## Evaluation of indels

With respect to indel length and position, mutations that were detected in the amplicon NGS reads were observed at positions ranging from 29 bp upstream to 54 bp downstream of the double stranded break (DSB+ (ORF nucleotide [nt] position -10 to +74)). Most indels were deletions of a single nucleotide, others were several nucleotides in length, and one of 62 nt was observed. Deletions were noted along the length of the amplicon, with several higher frequency sites indicated with bubbles of greater diameter in Fig 4C. The location of indels in the genomic DNAs pooled from juvenile flukes generally conformed with the location of programmed mutations detected in the genome of the adult stage liver flukes (Fig 4C, large bubbles). Insertions were seen only infrequently but these associated around these high frequency indel locations in juvenile and adult flukes. A cluster of mutations at nucleotide position -1 to -10 bp, within the 5' untranslated region (UTR) of the ORF, was notable given that 48 of the 216 indels (22%) observed in adult flukes occurred in this region, and five specific sites included indels detected in the genome of at least seven individual adult worms.

Fig 4D shows the frequency of mutations within each sample at each base pair position. There were positions in the ORF at which both control and *ΔOv-grn-1* juvenile and adults both exhibited a mutation, albeit at low frequency (<0.02%); a number/letter denotes *the location of these indels identify the nucleotide position and mutated residue*. The cluster of mutations noted above was situated within nucleotide positions -1 to -10 of the 5'-UTR of *Ov-grn-1* in adult and juvenile flukes. Similar alleles were not present in the control worms. Although the mutation rate in the 5'-UTR was low overall (<0.09%), numerous mutant alleles were seen at -9T and this single position comprised ~99% of the mutations for the $_{M/H}$*ΔOv-grn-1* adult and *ΔOv-grn-1* juvenile flukes.

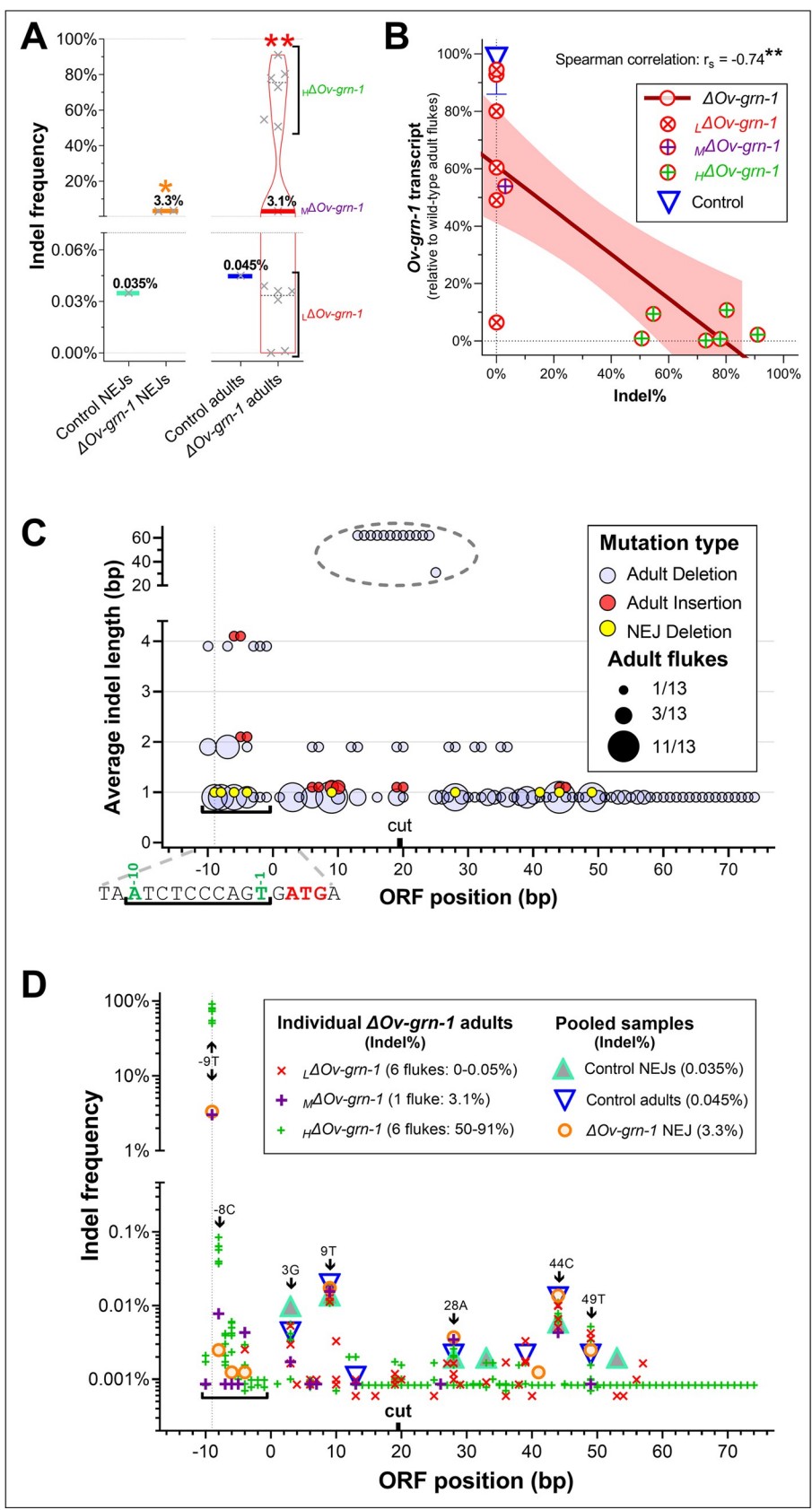

**Fig 4. Gene mutation rates among liver flukes.** Panel **A**, Programmed gene knock-out was highly efficient—although not in every fluke. The percentage of total indels (insertions/deletions) determined by next-generation sequencing was plotted on the vertical axis of juvenile and 24 wk adult flukes from Experiment 2. Control group juvenile (**NEJ**) and adult flukes were each from a single pooled sample while juvenile *ΔOv-grn-1* were from pools of two biological replicates, and *ΔOv-grn-1* adults from 13 individual flukes. The highly edited flukes were denoted $_H$*ΔOv-grn-1*, flukes with low editing denoted $_L$*ΔOv-grn-1*, and the single fluke with median level editing denoted $_M$*ΔOv-grn-1*. One sample t-test for either juvenile or adult worms comparing ΔOv-grn-1 and control: *, $P \leq 0.05$; **, $P \leq 0.01$. The thick solid line is the median and black dashed lines represent the inter-quartiles. A broken Y-axis with a magnified lower portion highlights the near zero values. **B.** Adult fluke indel mutation rate was inversely correlated with transcript level. The indel and transcript levels were plotted for each individual *ΔOv-grn-1* fluke (red circles, combining data from Figs 3C and 4A). Two-tailed non-parametric correlation determined by Spearman co-efficient: **, $P \leq 0.01$. For context, the control indel percentages were plotted against the transcript median (blue triangle) with interquartile range error bars. **C.** ΔOv-grn-1 indel location and size. The NGS reads revealed distinct indel patterns in 12 of 13 adult flukes. Shown as a multivariate bubble plot, the amplicon base pair open reading frame (ORF position) was plotted against the average indel length. The diameter of the bubbles (1–11) reflected how many of 13 adult flukes recorded a matching indel. The programmed double stranded break between residues 19 and 20 was indicated on the X-axis by the term "cut". For clarity, deletions in adult worm genomes (blue) and insertions (red) have been nudged up/down on the y-axis ±0.1. The deletions in juvenile worms (yellow) are shown from one pooled sample. Insertions were not seen. Position -9 was highlighted with a vertical dotted line and the black horizontal square bracket (⎵) highlighted a cluster of mutations. The sequence around this cluster was shown below the x-axis and the initiator ATG codon indicated in red. **D.** Mutation rate (indels) at each location: the graph plotted the nucleotide position against the percentage frequency of indels in individual flukes. A vertical dotted line highlighted a mutational hotspot at nucleotide -9 and the black horizontal bracket (⎵) marked a mutational cluster. Other indels of note were labeled with the nucleotide and position.

## Programmed knockout of *Ov-grn-1* impeded malignant transformation

Malignant transformation was apparent in hamsters by the end point of Experiment 2, at 24 weeks after infection. In the version of the model that we adapted [7], Syrian hamsters were infected with gene-edited juvenile *O. viverrini* flukes during concurrent exposure to exogenous nitrosamine (Fig 1A). At necropsy, prominent premalignant and malignant lesions were diagnosed in all three treatment groups of hamsters. Fig 5A and 5B present the gross anatomical appearance of livers from representative hamsters, where the CCA was visible to the unaided eye, highlighting the severity of disease manifested using this model. Specifically, multiple CCA nodules were obvious on both the diaphragmatic (Fig 5A) and visceral surfaces (Fig 5B). When micrographs of thin sections of the livers were examined, precancerous lesions were evident including biliary dysplasia (a precancerous precursor of CCA [30]) in many of the bile ducts and these were frequently accompanied by periductal fibrosis (Fig 5C). Fig 5D–5F presents representative photomicrographs from each of the treatment groups, highlighting the high-grade, malignant transformation, and Fig 5G and S2 Table summarize the findings from the treatment groups. Ten of 12 (83.3%) hamsters in the control group were diagnosed with CCA; high-grade CCA in eight and low-grade CCA in two. Dysplasia also was apparent in the remaining two of the 12 (one mild, one moderate) hamsters in the control group. CCA emerged in seven of 13 hamsters (six with high-grade CCA) in the *ΔOv-tsp-2* group. Of the remaining hamsters in this group, two showed mild biliary tract dysplasia, one showed biliary tract proliferation, two exhibited hepatobiliary inflammation, and a single hamster was free of apparent lesions. CAA was diagnosed in nine of 13 in the *ΔOv-grn-1* group, four of which showed high-grade CCA. Of the remaining hamsters, one showed dysplasia, two showed proliferation, and one was free of apparent lesions. Several hamsters infected with *ΔOv-tsp-2* (4/13 hamsters) and *ΔOv-grn-1* flukes (3/13 hamsters) exhibited lesions less severe than dysplasia, i.e. inflammation or proliferation, or were free of lesions.

Substantial differences were not evident among the treatment groups in the location or subtype of CCA tumors (S2 Table). Cholangiocarcinoma mass ranged from microscopic neoplasms with multifocal distributions (12/26) to tumor masses apparent to the unaided eye (14/26), with representative images in Fig 5A and 5B. With respect to histological classification,

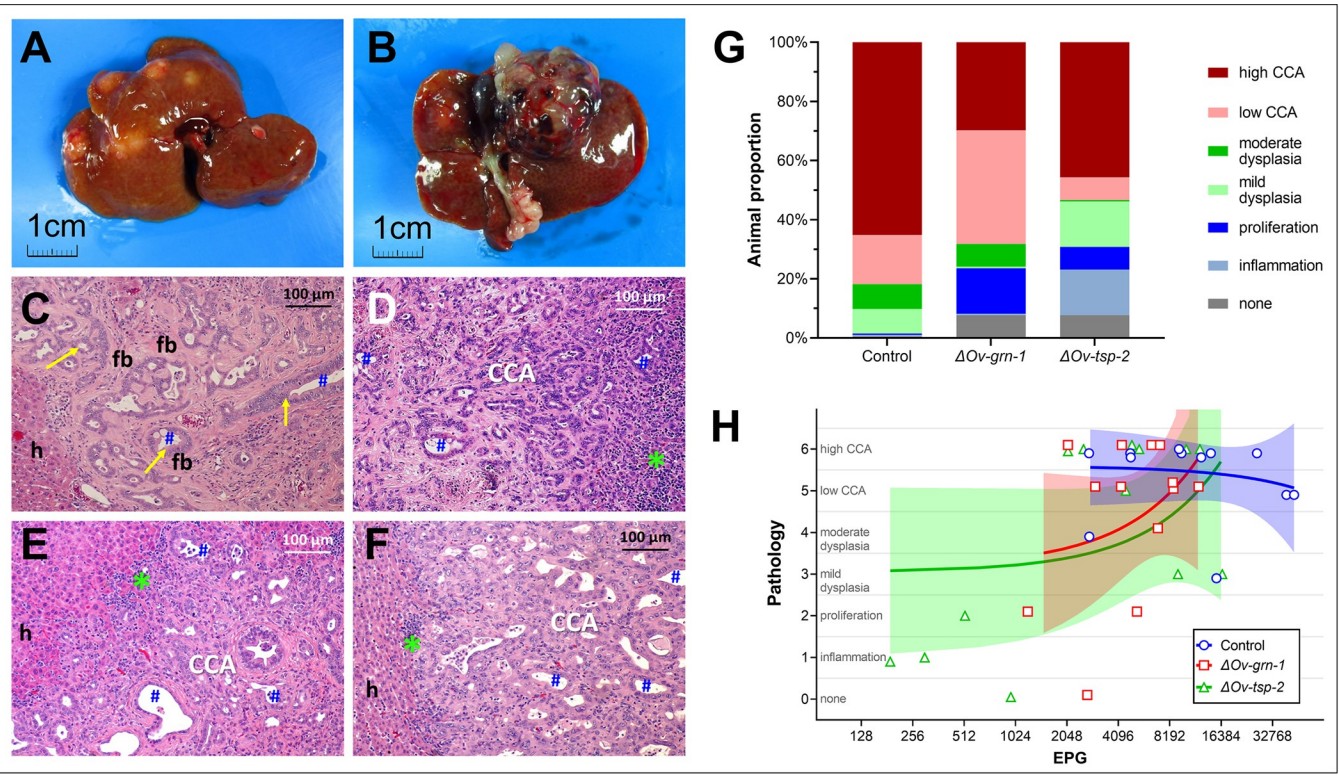

**Fig 5. Burden of disease in liver fluke infection-associated cholangiocarcinoma.** After resection of the livers at necropsy, a fragment of each lobe was either fixed in formalin for downstream thin sectioning or was manually disrupted to release flukes, which in turn were examined for gene editing events (Fig 1, Experiment 2). Gross anatomical appearance and histopathological results during induction of cholangiocarcinoma [31]. Multiple CCA nodules in the hamster liver were present on both diaphragmatic (**A**) and visceral surfaces (**B**). Micrographs of H&E-stained thin sections of liver highlighting foci of moderate dysplasia (**C**). This image shows bile ducts (blue #) encircled by dysplastic biliary epithelium (yellow arrow) surrounded by fibrosis (fb) with hepatocytes (h) to the left. H&E- stained images of CCA from each of the groups of hamsters group: control (**D**), *ΔOv-grn-1* (**E**), and *ΔOv-tsp-2* (**F**). Inflammation marked with green asterisk (*), cholangiocarcinoma labeled as CCA; other labels as in panel C. **G.** Assessment and scoring of lesions was undertaken independently by two co-authors (both veterinary pathologists) using anonymously labeled (blinded) micrographs. The severity of lesions increased from normal tissue (grey) to high grade CCA spanning multiple liver lobes (red). **H**. EPG from individual hamsters plotted against disease burden on a scale of zero (0, no lesion) to 6 (high CCA) scale. Data plots were slightly reformatted (nudged ± 0.1 on Y-axis) to enable display of overlapping points. Linear regression lines (which were not statistically significant) are shown in shaded color with 95% confidence intervals.

21/26 CCAs were the tubular type [1], and were seen in the three treatment groups. The papillary/cystic type was seen in a single instance, in an *ΔOv-grn-1* group hamster. Additionally, four mucinous type tumors were observed, one in the control and three in the *ΔOv-tsp-2* group hamsters. The right lobe was the common tumor location (20/26 livers), with nine in the left lobe, and three in the middle lobe. In some cases, tumors had developed in more than one lobe (5/26). Last, we assessed pathology in relation to EPG levels (Fig 5H) and noted there was less disease in *ΔOv-tsp-2* hamsters where EPG was lower, especially in hamsters with EPG < 1,000. However, significant correlation between pathogenesis and EPG was not apparent among the groups.

## Reduced fibrosis during infection with knockout parasites

Hepatic fibrosis was detected using Picro-Sirius Red (PSR)-stained thin tissue sections, which enabled investigation and quantification of development of peribiliary fibrosis (Fig 6A). Fibrosis was evaluated, firstly by Ishak staging, a semi-quantitative classification of the degree of fibrosis across the liver parenchyma [32] and, secondly, on the fibrotic deposition as

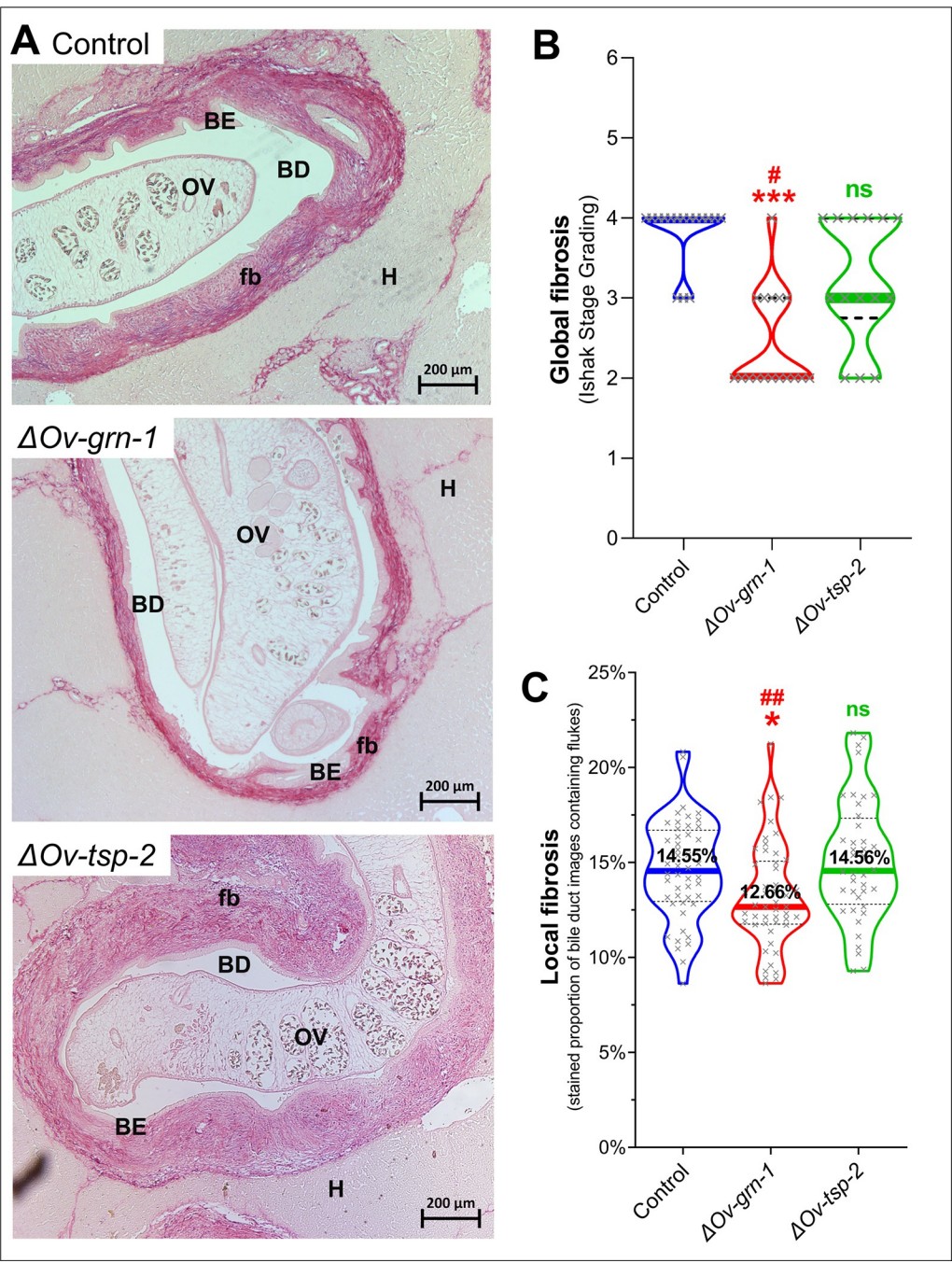

**Fig 6. Attenuated liver fibrosis during infection with ΔOv-grn-1 knockout parasites.** Panel **A.** Representative images of hepatic fibrosis stained by Picro-Sirius Red with CRISPR-Cas9 edited *O. viverrini*. Fibrosis was denoted as pink/red thick bands around the bile ducts (periductal fibrosis, fb) and expanded from each portal triad with fibrous septa. OV = *Opisthorchis viverrini*, H = hepatocyte, BD = bile duct, BE = biliary epithelium. **B**. Global liver fibrosis: Livers were scored for fibrosis using an Ishak Stage grading scale and plotted on the scale spanning from zero (no fibrosis) to six (cirrhosis) (n = 14 to 17 liver lobes per group).). **C.** Fibrosis proximal to flukes: Automated ImageJ fibrosis evaluation of the percentage of collagen deposition in images surrounding fluke-containing bile ducts (n = 46 to 54 images per group). Panels B, C: medians shown as thick colored line and dashed black lines mark the inter-quartile ranges. Kruskal-Wallis test with Dunn's multiple comparisons: against the control group, ns = not significant; *, P ≤ 0.05; ***, P ≤ 0.001, and against the ΔOv-tsp-2 group: #, P ≤ 0.05; ##, P ≤ 0.01.

demonstrated by staining with PSR localized around the liver flukes, i.e. the amount of collagen deposition surrounding bile ducts where individual liver flukes were situated at the time of necropsy.

The Ishak scores corresponded to degrees of injurious fibrosis and the levels reflect expansion of fibrosis into periportal regions and the degree of bridging between portal regions, culminating in frank cirrhosis which is scored as Ishak level 6. The control group was severely affected, with the majority (13/16) of the liver lobes assessed at Ishak score of 4 (median 4, range 3–4) (Fig 6B). Level 4 indicated that fibrosis had progressed extensively with marked portal-portal and portal-central bridging. Hamsters in the *ΔOv-grn-1* group showed only a single lobe graded at 4 and a majority of lobes (11/17) Ishak grade of 2 (median 2, range 2–4). Level 2 indicates less pathology, with fibrosis in most portal areas with/without short fibrous septa that had not bridged to other portal regions. Hamsters infected with the *ΔOv-tsp-2* liver flukes exhibited similar levels of disease among the hamsters in this group with Ishak scores ranging from 2 to 4 (median 3). Level 3 was defined as fibrous expansion of most portal areas with occasional bridging among them [32]. Periductal fibrosis of the hepatobiliary tract in the *ΔOv-tsp-2* group hamsters was not significantly different in Ishak score from the control group whereas the *ΔOv-grn-1* group displayed significantly less fibrosis than the control ($P \leq 0.001$) and *ΔOv-tsp-2* ($P \leq 0.05$) groups (Fig 6B).

Whereas the Ishak scores showed a significant difference between *ΔOv-grn-1* hamster livers and both other groups, data from Experiment 1 showed a higher worm burden in control animals than the gene-edited groups, and so the worm load may have contributed to greater fibrosis. There was no significant correlation between liver fibrosis and EPG for the control or *ΔOv-grn-1* fluke-infected groups, but a significant inverse correlation ($P \leq 0.05$) was detected for the *ΔOv-tsp-2* fluke-infected hamsters (S6 Fig). To minimize the impact on fibrosis of different worm burdens, we used automated image analysis to assess periductal fibrosis immediately proximal to liver flukes (which inhabit the lumen of the bile ducts). An ImageJ driven fibrosis quantification tool for PSR-stained collagen was deployed for automated analysis of the collagen deposition in periductal regions. Of the periductal regions proximal to flukes in the control and the *ΔOv-tsp-2* fluke-infected hamsters, median values of 14.55% (95% CI: 13.6–16.0) and 14.56% (95% CI: 13.5–16.0%) of fibrotic tissue were assigned, whereas significantly less (12.66%; 95% CI: 12.1–13.5%) bile duct tissue surrounding *ΔOv-grn-1* flukes was fibrotic (Fig 6C, $P \leq 0.05$). A concern remains over the influence of worm burden influencing pathology although both the control and the *ΔOv-tsp-2* groups showed similar localized collagen deposition factors despite dissimilar worm burdens. This outcome bolsters the use of an approach that focused on lesions immediately adjacent to detectable flukes as accurate and informative for comparison of pathogenesis on a per worm basis.

## *Ov-grn-1* KO flukes provoked less cell proliferation

Here we explored the *in vivo* effects of programmed knockout of *Ov-grn-1* expression, in contrast to earlier reports which dealt with proliferation of the biliary epithelium and/or cultured cholangiocytes in response to *in vitro* exposure to recombinant *Ov*-GRN-1 [14–17,33]. Proliferation of hamster biliary cells *in situ* was investigated using incorporation of a thymine analogue into cellular DNA. Visualization was performed using immunohistochemistry and evaluated quantitatively (Fig 7A). Concerning worm survival, and its corollary, the fitness cost of the programmed mutation, we examined proliferation in the bile ducts where flukes were situated. Median proliferation in bile duct tissue surrounding liver flukes in the control (15.0%, 95% CI: 12.4–22.8%) and *ΔOv-tsp-2* (11.1%, 95% CI: 4.1–20.0%) groups showed a wide range of values but overall these two groups were not significantly different from each

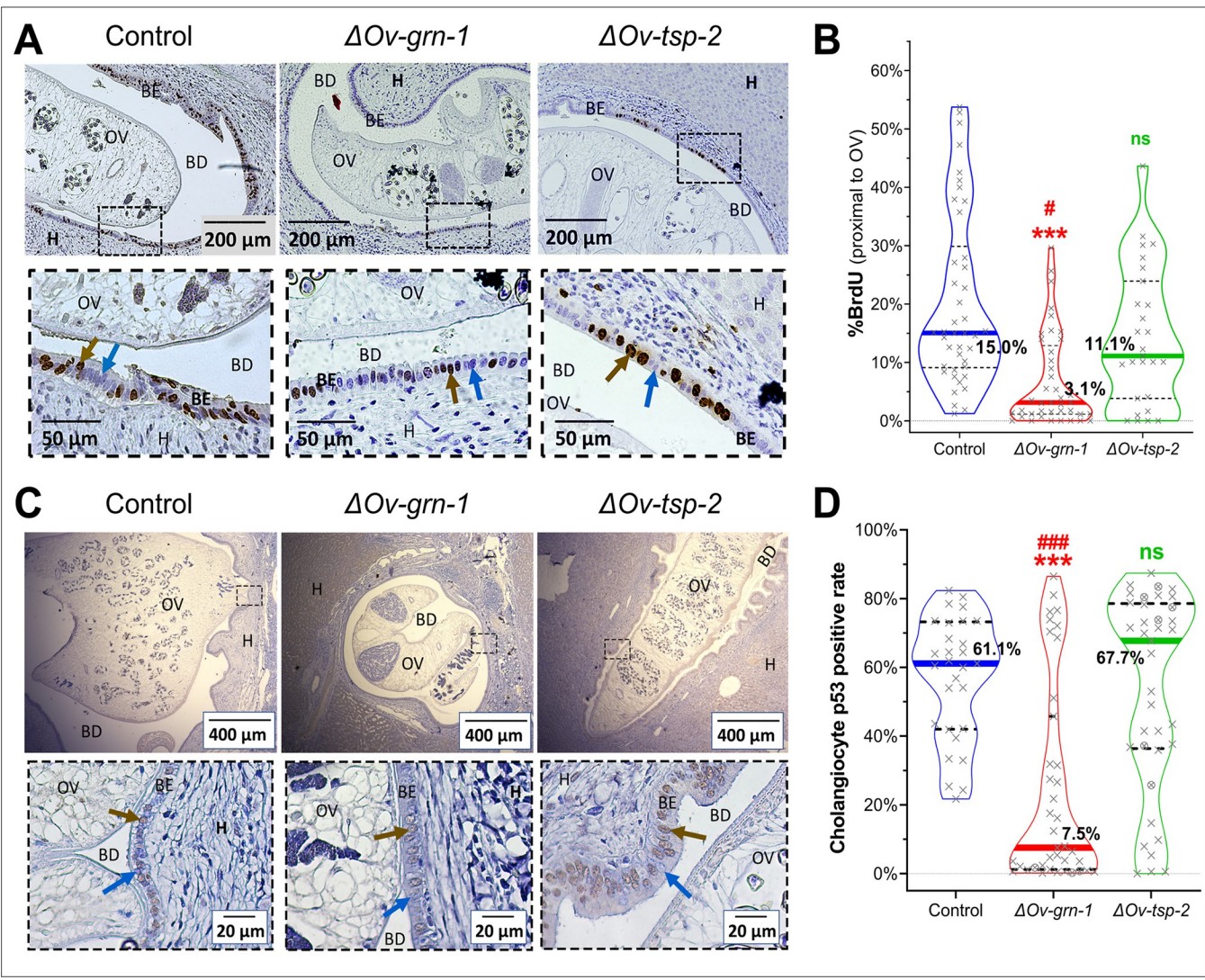

**Fig 7. Reduced proliferation and minimal mutant p53 expression in cholangiocytes in hamsters infected with *ΔOv-grn-1* genotype liver flukes.**
Representative images of biliary cells that incorporated BrdU from regions proximal to flukes in control, *ΔOv-grn-1* and *ΔOv-tsp-2* groups (**A**). The boxed region in the upper image is expanded in the lower panel. The brown arrow highlights the positive BrdU-stained nuclei and the blue arrow highlights a bile duct cell that did not incorporate BrdU. BrdU index measured from cholangiocytes adjacent to where a fluke was located (n = 27 to 42 per group). (**B**). Representative micrograph of p53 immunohistochemical staining of biliary epithelium during infection with gene edited flukes (**C**). Anti-mutant p53 antibody stained the nuclei brown (brown arrows); blue arrows indicate negative cells. Black dashed box in upper wide-angle image magnified in the lower image to aid visualization. Positivity rate (percentages) of mutant p53-positive cholangiocytes (n = 29 to 39 images per group) (**D**). Where available, 500 to 800 cells were scored from sections of each of the left, middle, and right lobes of the liver marked by "X". Fewer cholangiocytes (300–500) were available for assessment in several samples, denoted by ⊗ Panels A and C: OV = *Opisthorchis viverrini*, H = hepatocytes, BD = bile duct, BE = biliary epithelium. Panels B, D: non-parametric Kruskal-Wallis test with Dunn's multiple comparison correction compared against control: ns = not significant; ***, P ≤ 0.001, or against *ΔOv-tsp-2*: #, P ≤ 0.05; ###, P ≤ 0.001. Thick colored lines signify the median and the dashed black lines denote the inter-quartile range.

other (Fig 7B). The biliary epithelia surrounding *ΔOv-grn-1* flukes also showed substantial variation from 0–30% but the majority of the readings were below the other groups with a median value of 3.1% (95% CI: 1.6–7.6%) of cholangiocytes incorporating BrdU, which was significantly less proliferation than the control group (4.8-fold reduction, P ≤ 0.001) and the *ΔOv-tsp-2* group (3.6-fold reduction, P ≤ 0.05), respectively (Fig 7B).

## Mutant *TP53* less frequent during infection with *Ov-grn-1* KO flukes

Tumor protein p53 plays a well-recognized role in cholangiocarcinogenesis and *TP53* mutation is associated with fluke infection-associated malignancy [1,34]. A pattern of brown stained nuclei presented only in neoplastic biliary cells (Fig 7C). Wide angle images of flukes in the biliary tract showed mutant p53-positive and -negative cells in the epithelium. The profile of p53-positive cells differed markedly among the groups: control and *ΔOv-tsp-2* fluke-infected hamsters showed similar levels, 61.1% (95% CI: 43.6–68.3%) and 67.7% (95% CI: 41.5–73.5) of cholangiocytes stained for mutant p53, respectively (Fig 7D). By contrast, of the *ΔOv-grn-1* fluke-infected hamster bile ducts, 7.5% (95% CI: 2.3–27.5%) exhibited mutant p53-positive cells, ~13% of the levels that manifested in the other groups ($P \leq 0.01$) (Fig 7D).

## Discussion

CCA accounts for ~15% of all primary liver cancers globally and its incidence is increasing [35]. Infection with *O. viverrini* is the principal risk factor for CCA in the Lower Mekong River Basin countries including Thailand and Laos PDR, where CCA is the dominant form of liver cancer [1,22,36]. In an earlier report, we exploited this link to explore the role of *Ov*-GRN-1 secreted by the parasite in tumorigenesis using programmed gene knockout, and reported that the infection was less severe even though gene-edited parasites colonized the biliary tract of hamsters and developed into adult flukes [18]. In this follow-up investigation, we report findings during concurrent exposure to dietary nitrosamine and infection with the gene edited parasites, and that KO of the granulin gene retards malignant transformation to CCA, including the emergence of mutant p53, in a rodent model of human opisthorchiasis-associated CCA. These novel results build upon and advance the findings from our original report [18] and, notably, confirmed the role of liver fluke granulin in malignant transformation during chronic opisthorchiasis [15].

We utilized an established model of opisthorchiasis-associated CCA in hamsters that were infected with the parasite during concurrent exposure to exogenous nitrosamine. CCA manifests under these conditions, and this rodent model reflects the human situation where chronic opisthorchiasis in the context of a diet that is rich in fermented fish (in turn, rich in nitrosamines) culminates in a high incidence of CCA [20,22,37,38]. In hamsters, opisthorchiasis leads to periductal fibrosis. Chronic periductal fibrosis combined with a nitric oxide carcinogen, such as DMN, results in epithelial cholangiocyte proliferation, hyperplasia, dysplasia, and DNA damage, eventually and reliably manifesting as malignant neoplasia of the biliary tract [20,21,39]. By contrast, conspicuously less proliferation of the biliary epithelium, reduced mutant p53 expression by cholangiocytes, and less periductal fibrosis accompanied infection here with *ΔOv-grn-1* genotype worms compared to controls. As noted, our approaches and findings represent a functional genomics (reverse genetics)-focused extension of the model pioneered by Thai investigators more than 30 years ago [7].

Fitness cost of gene knockout can be assessed from programmed gene editing, an approach that is employed for the unbiased identification of essential genes in other organisms and disease settings [40,41]. The present findings confirmed the power of RNA-guided targeted mutation to define essentiality and relevance of two parasite proteins in infection-associated morbidity and malignancy. The *Ov-grn-1* gene does not appear to be essential for *in vivo* development and survival, which has enabled investigation here on the role of this protein in driving cell proliferation, pathology and ultimately contributing to CCA. Nonetheless, the reduced fecundity of *ΔOv-grn-1* liver flukes likely reflected a fitness deficit as the result of the targeted KO. By contrast, *Ov-tsp-2* appears to be essential to parasitism. The *ΔOv-tsp-2* genotype did not survive *in vivo*. These findings build upon earlier RNA interference-mediated silencing of

*Ov-tsp-2* gene expression and the resultant malformation of the tegument observed *in vitro* [42]. Although infection of hamsters with *Ov-tsp-2* dsRNA-treated parasites was not investigated in this earlier report, the damage to the tegument following exposure to *Ov-tsp-2* dsRNA in worms cultured for several days appeared to be so extensive and debilitating that worms damaged to that extent by either CRISPR-based genome knockout or dsRNA likely could not establish or survive for long periods *in vivo*.

In *O. viverrini*, *Ov*-GRN-1 and *Ov*-TSP-2 share key, though dissimilar functions at the parasite interface with the mammalian host. *Ov*-GRN-1 induces proliferation of cholangiocytes whereas *Ov*-TSP-2 is a key structural protein of the tegument of liver flukes (and indeed in schistosomes [43,44]) and of extracellular vesicles that are taken up by host cholangiocytes, among other roles [15,16,24]. Accordingly, *Ov-tsp-2* was included here as a comparator gene for *Ov-grn-1* KO. A non-targeting guide RNA encoded here by pCas-*Ov*-scramble, also was included, to provide a negative control for off-targeting by the Cas9 nuclease [26]. Ultimate lethality of *Ov-tsp-2* KO was borne out in our observation that *ΔOv-tsp-2* genotype flukes failed to survive to the adult stage in the hamsters. Whereas this diminished the value of *ΔOv-tsp-2* worms as controls, the findings highlighted the apparent essentiality of this tetraspanin to the intra-mammalian stages of the liver fluke. This essentiality of tetraspanin contrasted with *Ov*-GRN-1, the absence of which was not lethal to the parasite but which, fortunately, enabled inferences on its contribution to malignant transformation and which reinforced earlier hypotheses on its role as a carcinogen [15]. These approaches and findings are novel in the field of functional genomics for helminth parasites and hence, even if targeting *Ov-tsp-2*, in hindsight, was a sub-optimal choice as a control, the outcome provided a programmed mutation-based demonstration of the essentiality and lethality of mutations of these liver fluke genes, which represents vanguard progress in reverse genetics for helminth parasites. Indeed, for context concerning the pioneering significance of this advance, establishing the essentiality of human genes is an active and fertile field in functional genomics and gene therapy [45,46].

Notwithstanding that the primary goal of this investigation was to characterize pathogenesis and carcinogenesis associated with infection with *ΔOv-grn-1* flukes in this informative infection/nitroso-compound hamster model, we also investigated the mutation profile induced by the CRISPR-based targeted genome editing. Intriguingly, numerous mutations at the targeted *Ov-grn-1* locus were situated within the 5'UTR, rather than in the targeted exon, and most were detected at or proximal to a single residue, referred to here as -9T. This was situated 28 bp 5' to the programmed CRISPR/Cas9 double stranded break at nucleotide positions 19 and 20 of the open reading frame. This outcome was unforeseen given that mutations are usually expected at and adjacent to the programmed double stranded break, although guide RNAs exhibit individual, cell-line dependent biases toward particular outcomes [47]. The sequence of the 5'UTR of *Ov-grn-1* does not exhibit identity to regulatory elements (as identified using UTRScan [48]) which was unsurprising given that few helminth parasite UTR regulatory elements have been characterized [49]. Why this position was preferentially mutated is unclear although the marked reduction of transcription of *Ov-grn-1* that accompanied this mutation profile may signal the presence of a regulatory control element within the UTR.

Chronic inflammation and fibrosis are risk factors for liver cancer [50]. The traditional lifestyle of people living in *O. viverrini*-endemic areas, notably a diet enriched in raw fish and nitrosamines, as well as routine alcohol consumption, in tandem with the assault on the biliary epithelium by the attachment, feeding, movement, and secretions of the liver flukes that result in repeated cycles of injury and repair, establishes a compelling and conducive setting for malignant transformation [5,51,52]. The secretion of liver fluke granulin into the bile duct and

the ability of this growth factor to drive cell proliferation during infection and to (re)heal wounds inflicted by the parasite plays a central role in this process [15]. Whereas knockout mutation of *Ov-grn-1* did not prevent development and survival of the liver fluke *in vivo*, infection with these *ΔOv-grn-1* flukes failed to lead to marked cell proliferation and fibrosis in the immediate vicinity of the parasites, and consequently fewer hamsters developed high-grade CCA compared with hamsters infected with control and *ΔOv-tsp-2* parasites. Indeed, more hamsters infected with *ΔOv-grn-1* flukes were diagnosed in the categories termed either low-CCA and/or proliferation than in the other two groups. Knockout mutation of *Ov-grn-1* clearly impeded malignant transformation during chronic opisthorchiasis.

Infected hamsters exhibited elevated rates of *TP53* mutation, yet the level was markedly less during infection with *ΔOv-grn-1* flukes. The mutational signatures and related molecular pathways characteristic of human CCAs have been reviewed and the signature profiles differ between fluke-associated and non-fluke-associated CCAs [1,34]. Fluke-associated CCAs exhibit substantially more somatic mutations than non-fluke related CCAs [34], likely the consequence of opisthorchiasis-associated chronic inflammation. In conformity with the human situation, reduced inflammation and fibrosis were seen during infection with the *ΔOv-grn-1* flukes, further emphasizing the virulence of this growth factor in chronic opisthorchiasis. Inactivating mutations of *TP53* are more prevalent in CCA with a fluke infection etiology, as are mutations of *ARID1A*, *ARID2*, *BRCA1* and *BRCA2*, than in non-fluke related CCAs [34,53–56]. In addition, hypermethylation has been noted for the promoter CpG islands of several other aberrantly expressed genes [34].

Mosaicism of gene knock-out is a limitation of our somatic gene-editing approach. Obviating mosaicism by access to transgenic worms following germline transgenesis would clearly be preferable. However, access to transgenic lines of *O. viverrini*, while desirable, seems unlikely in the near future, especially considering the genetic complexity of this hermaphroditic platyhelminth obligate parasite with a diploid genome and a multiple host developmental cycle which cannot reliably be established in the laboratory [57]. Nonetheless, somatic genome editing is increasingly expedient in the clinic including for the treatment of hemoglobinopathies [58] and the identification of targets for disease intervention in translational medicine [59]. Given the role of liver fluke granulin as a virulence factor, the cogent link between CCA and liver fluke infection, and the dismal prospects following a diagnosis of CCA in resource poor settings [1,60,61], interventions that target this growth factor should be beneficial. Indeed, antibodies raised against liver fluke granulin block its ability to drive proliferation of CCA cell lines (14) and, hence, bootstrap support for a vaccination strategy targeting *Ov*-GRN-1 in the gastro-intestinal tract, for example through induction of mucosal IgA and IgG responses [23,62]. Such an intervention might contribute a productive component to a multivalent, orally administered, anti-infection and anti-cancer vaccine [1,23], which in turn would augment current tools for the public health intervention and control of this neglected tropical disease and its cancer burden [63,64].

## Materials and methods

### Ethics statement

The protocol for this research was approved by the Animal Ethics Committee of Khon Kaen University, approval numbers ACUC-KKU-61/60 and IACUC-KKU-92/63, which adhered to the guidelines prescribed by the National Research Council of Thailand for the Ethics of Animal Experimentation. All the hamsters were maintained at the animal husbandry facility of the Faculty of Medicine, Khon Kaen University, Khon Kaen.

## Metacercariae, newly excysted juvenile and adult developmental stages of *O. viverrini*

Metacercariae (MC) of *O. viverrini* were obtained from naturally infected cyprinid fish purchased from local food markets in the northeastern provinces of Thailand [65]. MC were isolated from fishes by using pepsin digestion as described previously [66]. Briefly, whole fishes were minced by electric blender and digested with 0.25% pepsin with 1.5% HCl in 0.85% NaCl at 37˚C for 120 min. The digested material was filtered sequentially through sieves of 1100, 350, 250, and 140 μm mesh apertures and the filtrate subjected to gravity sedimentation through several changes of 0.85% NaCl until the supernatant was clear. Sedimented MC were identified under a dissecting microscope as *O. viverrini*, and active (i.e., exhibiting larval movement within the cyst). MC were stored in 0.85% NaCl at 4˚C until used.

Newly excysted juveniles (NEJ) of *O. viverrini* were induced to escape from the metacercarial cyst by incubation in 0.25% trypsin in PBS supplemented with 2× 200 U/ml penicillin, 200 μg/ml streptomycin (2× Pen/Strep) for 5 min at 37˚C in 5% $CO_2$ in air. The juvenile flukes were isolated free of discarded cyst walls by mechanical passage through a 22 G needle [18]. We also use the term NEJ for the juvenile flukes because NEJ is also widely used for juveniles of related liver flukes [67,68].

## Plasmid constructs and transfection of *O. viverrini*

The CRISPR plasmid termed pCas-*Ov-grn-1*, encoding a guide RNA (gRNA) complimentary to *Ov-grn-1* exon 1 residues 1589–1608, 5'-GATTCATCTACAAGTGTTGA and encoding *Streptococcus pyogenes* Cas9, was designed using the online tools, http://crispr.mit.edu/ [69] and CHOPCHOP, http://chopchop.cbu.uib.no/ [70,71] using the *Ov-grn-1* gene (6,287 bp, GenBank FJ436341.1) as the reference, and constructed using the GeneArt CRISPR Nuclease Vector kit (Thermo Fisher Scientific), as described [18]. A second plasmid, termed pCas-*Ov-tsp-2* was constructed using the same approach; pCas-*Ov-tsp-2* encodes a gRNA, 5'-CTGGAACGTGGGCGAACCCG targeting exon 5 of the *Ov-tsp-2* gene (10,424 bp, GenBank JQ678707.1) [72,73]. The guide RNAs encoded by *Ov-grn-1* and *Ov-tsp-2* were predicted to exhibit high on-target efficiency and little or no off-target matches to the *O. viverrini* genome. A third construct, termed pCas-*Ov*-scramble was similarly prepared and included as a control to normalize analysis of gene expression and programmed gene knockout. The pCas-*Ov*-scramble construct included as the gRNA, a transcript of 20 nt, 5'- GCACTACCAGAGC-TAACTCA which exhibits only minimal identity to the *O. viverrini* genome [74]. A mammalian U6 promoter drives transcription of the gRNAs in all three plasmids and the CMV promoter drives expression of the *S. pyogenes* Cas9 nuclease, modified to include the eukaryotic nuclear localization signals 1 and 2. To confirm the orientation and sequences of gRNA in the plasmid vector, *Escherichia coli* competent cells (TOP10) were transformed with the plasmids, plasmid DNAs were recovered from ampicillin resistant colonies using a kit (Nucleo-Bond Xtra Midi, Macherey-Nagel GmbH, Düren, Germany), and the nucleotide sequences of each construct confirmed as correct by Sanger direct cycle sequencing using a U6-specific sequencing primer.

Two hundred juvenile *O. viverrini* were dispensed into an electroporation cuvette, 4 mm gap (Bio-Rad, Hercules, CA) containing 20 μg pCas-*Ov-grn-1*, pCas-*Ov-tsp-2* or pCas-*Ov*-scramble in a total volume of 500 μl RPMI, and subjected to a single square wave pulse at 125 V for 20 ms (Gene Pulser Xcell, Bio-Rad). These juvenile flukes were maintained in culture in RPMI supplemented with 1% glucose for 60 min after which they were used for infection of hamsters by stomach gavage (below).

## Infection of hamsters with gene edited *O. viverrini* juveniles

Wild-type (WT) flukes were collected and prepared for qPCR from infected hamsters 8 weeks after infection as previously described [18]. Fig 1 provides the timeframe and overview of the experimental designs employed in Experiments 1 and 2 including the infection concurrent with nitrosamine exposure used for the hamster CCA model. In Experiment 1, nine male hamsters (Syrian golden hamster, *Mesocricetus auratus*) aged between 6–8 weeks were randomly assigned into three experimental groups (Fig 1). Each hamster was infected with 100 *O. viverrini* NEJs by gastric gavage. These juvenile flukes had been transfected with pCas-*Ov-grn-1* plasmid, pCas-*Ov-tsp-2*, or the control pCas-*Ov*-scramble, and assigned the following identifiers: delta(Δ)-*gene* name, *ΔOv-grn-1*, *ΔOv-tsp-2*, or control, respectively. The infected hamsters were maintained under a standard light cycle (12 hours dark/light) with access to water and food *ad libitum*. Two weeks following infection, the drinking water provided to hamsters was supplemented with dimethylnitrosamine (DMN) (synonym, *N*-nitrosodimethylamine) (Sigma-Aldrich, Inc., St. Louis, MO) at 12.5 ppm until 10 weeks following infection [7,19,20]. Feces from each hamster were collected for fecal egg counts at weeks 10 and 12 after infection. The hamsters were euthanized at week 14 by inhalation overdose of isoflurane, followed by removal of the liver. Liver flukes were recovered from the liver, counted, and prepared for qPCR analysis of targeted genes. In Experiment 2, 45 male hamsters, 6–8 weeks of age, were randomly divided into three groups each with 15 hamsters and infected with gene edited NEJs and the drinking water supplemented with DMN (as in Experiment 1). Hamster feces were collected at week 23 after infection for fecal egg counts. At week 24, 40 mg/kg of the thymine analogue 5-bromo-2'-deoxyuridine (BrdU, Abcam, College Science Park, UK) was introduced into the peritoneum at 30 min before euthanasia, for investigation of proliferation of the biliary epithelia [75]. Hamsters were euthanized, the liver resected from each hamster, the liver lobes separated, and the left, middle and right lobes fixed in 10% formalin. We used a shorthand to label the lobes: left (left dorsocaudal), middle (combined ventral and dorsal median lobes), and right (right dorsocaudal). Liver flukes that were incidentally released from bile ducts during liver preparation and fixation for thin sectioning were retained and stored in RNAlater (Thermo Fisher). Levels of gene expression, mutation efficiency, and mutation profile were assessed in this sample of the liver fluke populations.

## Histopathological investigation

From each hamster, the entire liver was dissected and immersed in 10% buffered formalin. After overnight fixation, the liver lobes were processed for embedding in paraffin by dehydration through series of a 70%, 90%, and 100% ethanol, cleared in xylene, subjected to infiltration by paraffin at 56˚C, and last, embedding in paraffin. Four μm sections were sliced by microtome from the paraffin-embedded liver, and the sections stained with hematoxylin and eosin (H&E). Histopathological grading was undertaken by examination of the stained sections of liver and bile duct for inflammation, bile duct changes, dysplasia (including dysplasia in cholangiofibrosis), and stage of CCA as described [21,76–78], with modifications (Table 1).

## Fecal egg counts and worm counts

Feces from each hamster were individually collected, weighed and *O. viverrini* eggs per gram of feces (EPG) calculated using a modified formalin-ethyl acetate technique [82,83]. In brief, hamster feces were fixed in 10 ml of 10% formalin, the slurry of formalin-fixed feces filtered through two layers of gauze, and clarified by centrifugation at 500 *g* for 2 min. The pellet was resuspended with 7 ml of the 10% formalin, mixed with 3 ml ethyl-acetate and pelleted at 500 *g* for 5 min. The pellet was resuspended in 10% formalin solution and examined at 400× by light

**Table 1. Criteria for histopathological and histochemical assessment and grading.**

| Histopathological lesion | Grade description | Reference |
|---|---|---|
| Inflammation | 0 = None (no/minimal liver tissue or portal inflammation)<br>1 = Mild (1–2 foci per 4× objective at hepatocyte & periportal area)<br>2 = Moderate (3–5 foci per 4× objective at hepatocyte & periportal area)<br>3 = Severe (> 5 foci per 4× objective at hepatocyte & periportal area) | [21,78,79] |
| Bile duct changes | 0 = None (absence of proliferation and cholangiofibrosis)<br>1 = Mild (bile duct proliferation without cholangiofibrosis or periductal fibrosis)<br>2 = Moderate (bile duct proliferation with cholangiofibrosis)<br>3 = Severe (bile duct proliferation with cholangiofibrosis and periductal fibrosis) | [21,78,79] |
| Dysplasia | 0 = None (No cellular atypia, no nuclear polarity, no nuclear protrusions, no nuclear pseudostratification)<br>1 = Mild (Cellular atypia +ve, no nuclear polarity, no nuclear protrusions, nuclear pseudostratification +ve, nuclei within the lower two-thirds)<br>2 = Moderate (Cellular atypia +ve, nuclear polarity +ve, protruding of nuclei +ve, nuclear pseudostratification +ve)<br>3 = High (Cellular atypia ++ve, nuclear polarity ++ve, protruding of nuclei +ve, nuclear pseudostratification +ve) | [77] |
| Cholangiocarcinoma | 0 = None (no evidence of CCA)<br>Low CCA: 1 = Mild (CCA area 1–2 foci per 4× objective)<br>High CCA: combined 2+3: 2 = Moderate (CCA area 3–5 foci per 4× objective) and 3 = Severe (CCA area> 5 foci per 4× objective) | [21,78,79] |
| Fibrosis (PSR stain): Ishak score | 0 = No fibrosis<br>1 = Fibrous expansion of some portal areas, with or without short fibrous septa<br>2 = Fibrous expansion of most portal areas, with or without short fibrous septa<br>3 = Fibrous expansion of most portal areas with occasional portal to portal bridging<br>4 = Fibrous expansion of portal areas with marked bridging; portal to portal as well as portal to central<br>5 = Marked bridging (portal–portal and/or portal–central) with occasional nodules (incomplete cirrhosis)<br>6 = Cirrhosis, probable or definite | [32,80] |
| Assessment of collagen proximal to liver flukes | Quantitative automated evaluation of collagen deposition percentage surrounding bile ducts | ImageJ MRI Fibrosis Tool, developed by Volker Bäcker, https://github.com/MontpellierRessourcesImagerie/imagej_macros_and_scripts/wiki/MRI_Fibrosis_Tool, [18,81] |

microscopy. EPG was calculated as follows: (average number eggs × total drops of fecal solution)/ gram of feces. At necropsy, intact mature *O. viverrini* from the hepatobiliary tract were recovered during observation of the livers using a stereo dissecting microscope and stored for downstream investigation of programmed mutation.

## Extraction of nucleic acids

Pooled NEJ or single mature worms were homogenized in RNAzol RT (Molecular Research Center, Inc., Cincinnati, OH) before dual RNA and DNA extraction, as described [18]. Briefly, the parasite(s) were homogenized in RNAzol RT using a motorized pestle, after which the DNA and protein were precipitated in nuclease free water. The aqueous upper phase was transferred into a new tube for total RNA precipitation by isopropanol (50% v/v). The DNA/protein pellet was resuspended in DNAzol and genomic DNA extracted according to the manufacturer's instructions (Molecular Research Center). Concentration and integrity of genomic

DNA and total RNA were quantified by spectrophotometry (NanoDrop 1000, Thermo Fisher, Waltham, MA). Transcription and expression were investigated in pools of NEJs and in individual adult flukes after normalization against the controls.

## Quantitative real-time PCR

cDNA was synthesized from DNase I-treated-total RNA (10 ng) using the Maxima First Strand cDNA synthesis with a DNase kit (Thermo Scientific) prior to performing quantitative real-time PCR (qPCR). Each cDNA sample was prepared for qPCR in triplicate using SSoAdvanced Universal SYBR Green Supermix (Bio-Rad). Each qPCR reaction consisted of 5 μl SYBR Green Supermix, 0.2 μl (10 μM) each of specific forward and reverse primers for *Ov-grn-1* (forward primer, *Ov*-GRN-1-RT-F: 5'-GACTTGTTGTCGCGGCTTAC-3' and reverse primer, *Ov*-GRN1-RT-R: 5'-CGCGAAAGTAGCTTGTGGTC-3'), amplifying 147 base pairs (bp) of 444 nt of *Ov-grn-1* mRNA, complete cds GenBank FJ436341.1) or primers for *Ov-tsp-2* (forward primer, *Ov-TSP-2*-F 5'- ACAAGTCGTATGTGGAATCA- 3' and reverse primer *Ov-TSP-2*-R 5'- CCGTCTCGCCTTCTCCTTT- 3', product size 377 bp of 672 nt of *Ov-tsp-2*A mRNA, complete cds (GenBank JQ678707.1), 2 μl cDNA and distilled water to a final reaction volume of 10 μl. The thermal cycling included an initiation cycle at 95˚C for 3 min followed by 40 cycles of denaturation at 95˚C for 15s and annealing at 55˚C for 30s using CFX Connect Real-Time PCR system (Bio-Rad). The endogenous actin gene (1301 nt of *Ov-actin* mRNA, GenBank EL620339.1) was used as a reference [17,84,85] (forward primer, *Ov*-actin-F: 5'-AGCCAACCGAGAGAAGATGA and reverse primer, *Ov*-actin-R: 5'-ACCTGACCATCAGG-CAGTTC. The fold change in *Ov-grn-1* and *Ov-tsp*-2 transcripts was calculated using the $2^{(-\Delta\Delta Ct)}$ method using the *Ov-actin* gene as a reference for normalization [17,84,85]. Transcript ddCT qPCR data were resampled with replacement bootstrap analysis in Microsoft Excel with 1000 bootstrap resamples ($B$ = 1000, n = original sample number) to generate mean values and 95% confidence intervals [86].

## Illumina based targeted next generation sequencing of targeted amplicons

The Amplicon-EZ next generation sequencing approach (GENEWIZ, South Plainfield, NJ) was used to obtain deeper coverage of *Ov-grn-1* exon 1 from individual mature worms or pooled NEJ, which provided > 50,000 complete amplicon aligned read-pairs per sample. A 173-nucleotide region flanking the programmed DSB was amplified with forward primer 5'-TTCGAGATTCGGTCAGCCG-3' and reverse primer 5'-GCACCAACTCGCAACTTACA-3', and amplicons sequenced directly using Illumina chemistry. The CRISPR RGEN Tool web platform (http://www.rgenome.net/about/), comparison range 60 nt, was used to screen for Cas9-catalyzed substitutions in the aligned read-pairs with comparisons among the treatment groups [29,87]. In addition, CRISPResso2 with a quantification window set at 30 nt [28] was employed for indel estimation, as described [18]. The sequence reads are available at GenBank BioProject PRJNA385864, Sequence Read Archive study SRP110673, sequence runs SRR15906234-15906251, accessions SRX12196673-SRX12196690.

## BrdU-staining for proliferation of the biliary epithelium

Proliferation of biliary epithelial cells was investigated by using incorporation of BrdU. In brief, the liver sections of a paraffin-embedded sample were soaked in xylene, rehydrated in graded alcohol solution (100%, 90%, and 70% ethanol for 5 min each), and antigen was retrieved in citrate buffer (pH 6) for 5 min in a pressure cooker. The tissue sections were blocked with 3% $H_2O_2$ in methanol for 30 min and subsequently incubated with 5% fetal bovine serum in phosphate buffered saline for 30 min at room temperature (RT). The sections

were incubated with mouse anti-BrdU monoclonal antibody (Abcam, catalogue no. ab8955) diluted 1:200 in PBS at 4°C overnight, and then probed with goat anti-mouse IgG-HRP (Invitrogen, Thermo Fisher) diluted 1:1,000 in PBS for 60 min at RT. The peroxidase reaction was developed with 3, 3'- diaminobenzidine (DAB). Sections were counterstained with Mayer's hematoxylin for 5 min before dehydrating and mounting. A positive signal was indicated by a brown colored precipitate under light microscopy. I mages were captured with a Zeiss Axiocam camera ICc5 and NIS-Element software (Nikon, Minato, Tokyo, Japan). To quantify BrdU-positive nuclei, cholangiocytes were counted in 10 non-overlapping fields of 400x magnification, with a total of 1,000 biliary cholangiocytes counted using the counter plug-in tool at ImageJ 1.52P. The level of cell proliferation was quantified as a percentage using the formula: positive biliary nuclei/total biliary cells x 100%.

## Staining for mutant forms of p53

To investigate levels of p53 mutation [88] in cholangiocytes, paraffin-embedded tissue sections were deparaffinized and rehydrated by standard methods. Thereafter, sections were incubated with monoclonal mouse anti-p53 (mutant, clone Ab-3 PAb240 catalogue no. OP29-200UG) (Merck, Darmstadt, Germany) diluted 1:100 in PBS at 4°C overnight, and after thorough washing, probed with goat anti-mouse IgG-HRP (Invitrogen, Carlsbad, CA) diluted 1:1,000 in PBS for 60 min at 25°C. The peroxidase reaction was developed with 3,3'- DAB and sections counterstained with Mayer's hematoxylin for 5 min. A human CCA cell line served as the positive control for mutant p53 positivity [89,90]. Images of high-power fields (400x magnification) of the biliary epithelium were captured in five non-overlapping fields of each of the right, middle, and left liver lobes using a Zeiss Axiocam fitted with a ICc5 camera and NIS-Element software (Nikon). The percentage of mutant p53-positive cholangiocytes was determined by scoring positive cells from 500 to 800 cholangiocytes from the right, middle, and left lobes of the liver using imageJ.

## PSR staining to evaluate fibrosis

Liver tissue sections containing resident *O. viverrini* were selected for fibrosis measurements using PSR staining for collagen (Abcam, catalogue ab150681). The thin sections were deparaffinized in xylene and rehydrated through an ethanol gradient. The sections were immersed in the PSR solution and incubated at 25°C for 60 min. Residual dye was removed by washing twice in dilute acetic acid (0.5%) after which sections were dehydrated through graded series of ethanol and xylene, the slides cleared with 100% xylene, mounted in Per-mount, and air dried overnight. Fibrosis surrounding the bile duct (periductal fibrosis, PF) proximal to the liver flukes was evaluated by two approaches: First, using scoring according to accepted criteria [32], samples were blinded and fibrosis scores (0–6) were assigned semi-quantitatively using the Ishak stage (Table 1) by two experienced pathologists; and, second, localized fibrosis was evaluated by capturing images for quantification of collagen deposition. Specifically, the PSR-stained fibrotic lesions were measured using the plug-in fibrosis tool developed by Volker Bäcker (New FUJI toolsets for bioimage analysis, available at https://github.com/ MontpellierRessourcesImagerie/imagej_macros_and_scripts/wiki/MRI_Fibrosis_Tool). We [18] and others [81,91,92] have previously used this tool with PSR-stained tissues.

## Statistical analysis

One-way ANOVA with Tukey multiple comparisons was used for comparisons with two to four replicates (worm burden and EPG at week 14). The Krustal-Wallis non parametric test with Dunn's multiple comparisons was used for datasets that were not normally distributed

including the EPG values at 24 weeks, Ishak and periductal fibrosis, BrdU scores, and mutant p53 signals. Numbers of replicates and errors bars are provided in the figure legends. Statistical analysis and graphical presentation of the results were undertaken using GraphPad Prism version 9 (GraphPad Software Inc, San Diego, CA). To compare mutation rate, a one sample *t*-test compared the replicate % indel values for the *ΔOv-grn-1* against the single control group % indel for either juvenile or adult *O. viverrini* worms. Assessment for correlation between % indel and transcription was carried out using a two-tailed non-parametric Spearman correlation co-efficient ($r_s$). Values of $P \leq 0.05$ were considered to be statistically significant: an asterisk (*) corresponds to the control vs *ΔOv-grn-1* group comparison; *, $P \leq 0.05$, **, $P \leq 0.01$, ***, $P \leq 0.001$, hashtag (#) for *ΔOv-grn-1* vs *ΔOv-tsp-2* group comparison; #, $P \leq 0.05$; ##, $P \leq 0.01$; ###, $P \leq 0.001$.

## Supporting information

**S1 Fig. Transcript levels of gene edited NEJ flukes with bootstrapped population values.** Each group of flukes was subjected to gene editing targeting *Ov-grn-1* (*ΔOv-grn-1* flukes), *Ov-tsp-2* (*ΔOv-tsp-2 flukes*), or with an irrelevant guide RNA as a control (Control). Relative transcript levels were plotted for both the *Ov-grn-1* (A) and *Ov-tsp-2* genes (B) for all three groups: control, ΔOv-grn-1, and ΔOv-tsp-2 flukes. Each panel shows ddCt (delta-delta cycle threshold) biological replicate values plotted relative to newly excysted juvenile (NEJ) control average. Resampling with replacement bootstrap analysis (B = 1000) of ddCT scores used to generate population average denoted by thick colored line and 95% confidence interval bars. (DOCX)

**S2 Fig. Experiment 1: Relationship of fecal egg count to worm burden.** Assessment of eggs per gram of feces (EPG) at 10 (**A**) and 12 (**B**) weeks after infection compared to worm burden at necropsy (week 14). The worm burden for each hamster was plotted against the EPG at weeks 10 and 12. Comparing these timepoints (**C**) in a linear regression analysis did not reveal variation from a line with zero slope (horizontal line) at either interval. Each hamster was designated by letter, C = control, G = *ΔOv-grn-1*, T = *ΔOv-tsp-2*, with the number of the hamster, as 1, 2, and 3. (DOCX)

**S3 Fig. Experiment 1: Transcript levels of gene edited adult flukes with bootstrapped population values.** Each group of the flukes was subjected to gene editing targeting *Ov-grn-1* (*ΔOv-grn-1* flukes), *Ov-tsp-2* (*ΔOv-tsp-2* flukes), or with an irrelevant guide RNA as a control (Control). Each panel shows ddCt (delta-delta cycle threshold) of individual flukes plotted relative to transcript levels of wild-type flukes for *Ov-grn-1* (**A**) and *Ov-tsp-2* (**B**). The dashed line purple box inset is an enlarged region of panel B, included for clarity. Resampling with replacement bootstrap analysis (B = 1000) of ddCT scores used to generate population average denoted by thick colored line and 95% confidence interval bars. (DOCX)

**S4 Fig. Transcript levels of adult flukes from Experiment 2 with bootstrapped population values.** Each group of the 24-wk-old flukes was subjected to gene editing targeting *Ov-grn-1* (*ΔOv-grn-1* flukes), Ov-tsp-2 (*ΔOv-tsp-2* flukes), or with an irrelevant guide RNA as a control (Control). Each panel shows ddCt (delta-delta cycle threshold) for individual flukes plotted relative to wild-type (WT) fluke transcript levels for *both Ov-grn-1* (A) and *Ov-tsp-2* genes (B). The dashed line purple box inset is an enlarged section of panel B for clarity. Resampling with replacement bootstrap analysis (B = 1000) of ddCT scores used to generate population average

denoted by thick colored line and 95% confidence interval bars.
(DOCX)

**S5 Fig. Profiles of nucleotide substitutions.** Nucleotide substitution profiles detected in the 173 bp amplicon spanning the programmed cleavage site in *Ov-grn-1* from both juvenile (NEJ) and single adult *O. viverrini* flukes of the *ΔOv-grn-1* treatment group compared with the irrelevant guide RNA-treated control group. RGEN's CRISPR-sub analysis tool, http://www.rgenome.net/crispr-sub/#!, aligns read-pairs to plot the substitution patterns among Illumina sequence reads from amplicon libraries from CRISPR/Cas9 editing-focused datasets. Experimental group (red, upper axis) versus control group (blue, lower axis); the X-axis shows the targeted gene including programmed cleavage site (position 0) between nucleotides 19 and 20 of ORF 1 of *Ov-grn-1*. Juvenile flukes are shown in the top left and each adult fluke is shown separately and designated with the number of its host hamster number (x) 1–15 and worm number (y) 1–3: (h"x" worm"y"). Substantial differences were not apparent in patterns of substitutions detected among the experimental and control groups.
(DOCX)

**S6 Fig. Correlation between fecal EPG and fibrosis.** For Experiment 2, each liver lobe was plotted as eggs per gram against the Ishak fibrosis (data combined from Figs 3A and 6B). Data points have been nudged ± 0.1 on the vertical axis for clarity among overlapping points. The linear regression line for each group is shown; ns, not significant; *, $P \leq 0.05$.
(DOCX)

**S1 Table. Summary of NGS data.** Frequency of successful CRISPR/Cas9 gene knock out editing as determined by frameshift mutations. The CRISPResso2 analysis used a window size (-3 option) that include the entire 173 bp amplicon, except for 25 bp at each end in order to the exclude the primer binding regions. Combined insertions and deletions are used to generate the % indel, the proportion of read pairs aligned (RPA). To match the color scheme in the figures, the control groups highlighted in blue and low, medium and highly edited adults are highlighted in red, purple and green. Most indels are located in the 5' UTR and are not frameshift mutations.
(DOCX)

**S2 Table. Assessment of pre-malignant and malignant lesions.** Disease profiles among the three groups of hamsters and the histopathological diagnosis data was summarized as graphs in Fig 5. Beyond Fig 5, the table outlines the histopathological type of the cholangiocarcinoma (CAA), location, and tumor progression. Representative images of the three major CCA types, **A** = tubular, **B** = papillary, and **C** = mucinous, are included.
(DOCX)

## Acknowledgments

We thank Suwit Balthaisong for technical assistance.

## Author Contributions

**Conceptualization:** Sujittra Chaiyadet, Sirikachorn Tangkawattana, Michael J. Smout, Wannaporn Ittiprasert, Patpicha Arunsan, Alex Loukas, Paul J. Brindley, Thewarach Laha.

**Data curation:** Michael J. Smout, Wannaporn Ittiprasert, Paul J. Brindley.

**Formal analysis:** Sirikachorn Tangkawattana, Michael J. Smout, Wannaporn Ittiprasert.

**Funding acquisition:** Alex Loukas, Paul J. Brindley, Thewarach Laha.

**Investigation:** Sujittra Chaiyadet, Sirikachorn Tangkawattana, Michael J. Smout, Wannaporn Ittiprasert, Victoria H. Mann, Raksawan Deenonpoe, Patpicha Arunsan.

**Methodology:** Sujittra Chaiyadet, Sirikachorn Tangkawattana, Michael J. Smout, Wannaporn Ittiprasert, Raksawan Deenonpoe.

**Project administration:** Alex Loukas, Paul J. Brindley, Thewarach Laha.

**Resources:** Michael J. Smout, Wannaporn Ittiprasert, Alex Loukas, Paul J. Brindley, Thewarach Laha.

**Software:** Michael J. Smout, Wannaporn Ittiprasert.

**Supervision:** Alex Loukas, Paul J. Brindley, Thewarach Laha.

**Validation:** Sirikachorn Tangkawattana, Wannaporn Ittiprasert, Alex Loukas, Paul J. Brindley, Thewarach Laha.

**Visualization:** Sujittra Chaiyadet, Michael J. Smout, Wannaporn Ittiprasert.

**Writing – original draft:** Sujittra Chaiyadet, Sirikachorn Tangkawattana, Michael J. Smout, Wannaporn Ittiprasert, Victoria H. Mann, Alex Loukas, Paul J. Brindley, Thewarach Laha.

**Writing – review & editing:** Michael J. Smout, Victoria H. Mann, Alex Loukas, Paul J. Brindley, Thewarach Laha.

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
