## [Decision Letter · Decision Letter 0]

31 May 2022

Dear Dr. Brindley,

Thank you very much for submitting your manuscript "Programmed knockout mutation of liver fluke granulin, Ov - grn - 1 , impedes malignant transformation during chronic opisthorchiasis" for consideration at PLOS Pathogens. As with all papers reviewed by the journal, your manuscript was reviewed by members of the editorial board and by several independent reviewers. The reviewers appreciated the attention to an important topic. Based on the reviews, we are likely to accept this manuscript for publication, providing that you modify the manuscript according to the review recommendations.

In our opinion, the revision does not require performance of further experiments. Specifically, Reviewer 2 suggested but did not require further experimental work concerning loss of an off-target gene disruption control, but also accepted that the question can be afddresed by modification of the text.

Sincerely,

Kenneth D Vernick

Associate Editor

PLOS Pathogens

P'ng Loke

Section Editor

PLOS Pathogens

Kasturi Haldar

Editor-in-Chief

PLOS Pathogens

orcid.org/0000-0001-5065-158X

Michael Malim

Editor-in-Chief

PLOS Pathogens

orcid.org/0000-0002-7699-2064

In our opinion, the revision does not require performance of further experiments. Specifically, Reviewer 2 suggested but did not require further experimental work concerning loss of an off-target gene disruption control, but also accepted that the question can be afddresed by modification of the text.

Reviewer Comments (if any, and for reference):

Reviewer's Responses to Questions

**Part I - Summary**

Reviewer #1: In this manuscript by Chaiyadet et.al. the investigators used a CRISPR/Cas9 system to knockout a gene in the liver fluke Opisthorchis viverrine and showed that the Ov-GRN-1 gene is important for disease pathogenesis. It should be noted that the ability to genetically manipulate helminth parasites to study pathogenesis has been one of the main limiting factors in the field of helminth biology and hence this represents an important breakthrough for the field, especially for the study of this parasite. While the efficiency of deletion is not ideal, for example in Figure 2 C, there is clearly a bimodal distribution to deletion, with half the flukes having no deletion and the other half having good deletion. Most importantly, the investigators were able to demonstrate that edited parasites had a consequence on fibrosis in an animal model. Mechanistically, this was associated with reduced p53 expression and proliferation of Cholangiocytes, which are the targets for this molecule. This is an important manuscript and a remarkable piece of work by the investigators that is going to have a very high impact on the field of helminth biology.

Reviewer #2: Review of MS# PPATHOGENS-D-22-00097 by Chaiyadet, Loukas, Brindley, Laha and others entitled “Programmed knockout mutation of liver fluke granulin, Ov-grn-1, impedes malignant transformation during chronic opisthorchiasis”.

This paper is a continuation of ongoing study by the authors of the role of the liver fluke Opisthorchis viverrini in the pathogenesis of human cholangiocarcinoma. Association of O. viverrini infection with this virulent cancer is clear, and preceding studies by this group demonstrated that a secreted tissue repair factor, the granulin encoded in Ov-grn-1, accounts for the majority of the worm’s pathogenic effects on the hepatobiliary tract. The most recent findings revealed that disruption of Ov-grn-1 by CRISPR/Cas9, a method developed for parasitic flatworms by the authors, resulted in strong suppression of the gene’s transcripts and, concomitantly, of pathologic changes in hepatobiliary tissues surrounding gene-edited worms. This study recapitulates these findings in a study that introduces chronic nitrosamine exposure in the infected hamster model to mimic the contribution of these compounds that would accompany O. viverrini infection by ingestion of preserved fish intermediate hosts. Additionally, the authors demonstrated that specific indicators of malignant transformation (TP53 mutation and cell proliferation) that are upregulated in infections with wild-type parasites, are suppressed by disruption of Ov-grn-1. These are noteworthy additions to the body of knowledge about O. viverrini infection and cancer.

This reviewer notes some substantive shortcomings of the paper, two of which could be dealt with by careful revision of the text. One other, the essential loss of an off-target gene disruption control, may require further experimentation, but at least should be discussed thoroughly by the authors within a revised MS. There are a few minor points of grammar and editing listed as well.

Reviewer #3: This study provides further evidence that the liver fluke protein granulin is involved in the causative pathway to mutagenesis and resulting tumour formation in hamster models. Overall the experiments performed appear to be appropriate and the findings, as claimed, present an advance on current studies. However I raise a number of issues with the presentation of data and the statistical tests performed, which makes it difficult to interpret the study results and assess whether the claimed findings are justifiable.

**Part II – Major Issues: Key Experiments Required for Acceptance**

Reviewer #1: (No Response)

Reviewer #2: SUBSTANTIVE POINTS

Note: Of these, only point #2 may require additional experimentation, that is assessment of pathogenesis by worms with an off-target gene knockout that allows development of NEJ parasites to the adult stage. Alternatively the authors should thoroughly consider and discuss in the paper the implications of the loss of the Ov-tsp-2 edited controls.

1. Line 73: readers may be struck by the similarity of this study and its findings to one reported in the 2019 Elife paper (reference 7). It would be helpful at some point, preferably early in the paper, to explicitly state the novel aspects of this study in relation to that previous one. This reviewer assumes the concurrent treatment with nitrosamine is one such additional measure. Is that correct? Also, assessments of cell proliferation and emergence of specific markers of malignant transformation (TP53) appear novel. Also correct? Please elaborate on these and other points to clearly differentiate this paper from the 2019 publication.

2. Line 137 and in general: it’s not clear to this reviewer why Ov-tsp-2 was selected as a control gene for knockout. Was it simply a non-target gene against which to compare phenotypes resulting from OV-tsp-2 knockout? If so, it’s unclear why a gene that’s essential for normal development in other parasitic flatworms, such as schstosomes, would be selected. Ultimate lethality of Ov-tsp-2 knockout was borne out by the data indicating that no or few Ov-tsp-2 knockout worms survived to the adult stage in the infected hamsters. That would seem to diminish or eliminate the value of these Ov-tsp-2 knockouts as controls. The authors should discuss this point, evaluating the impact of losing this control on the integrity of the study.

3. Line 536-543: Given this observation, readers may question the advantages of targeted gene mutation via CRISPR/Cas9 over previous RNAi based approaches to functional genomics in parasitic flatworms, which also resulted in significant, but not complete ablation of target gene transcripts and, in studies of putative virulence factors, incomplete resolution of lesions. Can the authors comment on this in light of previously published work?

Reviewer #3: None

**Part III – Minor Issues: Editorial and Data Presentation Modifications**

Reviewer #1: (No Response)

Reviewer #2: MINOR POINTS OF EDITING AND GRAMMAR

4. Line 40: suggest inserting “(CCA)” after “cholangiocarcinoma”.

5. Lines 70-71: consider "diseases of the liver and bile ducts" rather than "hepatobiliary diseases" for lay readers.

6. Line 91: this sentence might read more smoothly if you made it “….CCA are…” rather than “…CCA is..”.

7. Line 105: consider “promotes” rather than “can promote”.

8. Line 109-110: suggest deleting “, shown” after “is feasible and”.

9. Line 113: perhaps "the pathogenesis of CCA", rather than “induction of CCA”?

10. Line 119-123: this is a wordy and cumbersome sentence as written. Consider revising for conciseness and clarity.

11. Line 128: consider “highlighted” rather than “highlight”.

12. Line 187: substitute “did not show” for “not showing”.

13. Line 188: likewise, substitute “showed” for “showing”.

14. Line 223: make it “fecal egg counts”.

4. Line 286: delete “a” after “showed”.

5. Line 388: need a comma after “infection”.

6. Line 413: make it "different from the control group".

7. Line 426: suggest deleting “as a consequence”.

8. Line 621: The designation Syrian golden hamsters, Mesocricetus auratus should be moved to the first reference to hamsters in Materials and Methods.

9. Line 663: Suggest “Feces from each hamster were…”.

10. Line 793: Shouldn’t it read “P<0.05”?

11. Line 833 and following: see to italicizing genus/species names in the bibliography.

Reviewer #3: These comments are given in the order at which they appear in the text, rather than in order of importance:

1. The article title should be revised, "programmed knockout mutation" seems redundant and is identical to the title of a related article by the same authors; "Programmed knockout mutation of liver fluke granulin attenuates virulence of infection-induced hepatobiliary morbidity" (Arunsan 2019 Elife). Consider instead "Knockout of liver fluke granulin, Ov- grn-1 , impedes

malignant transformation during chronic infection with Opisthorchis viverrini"

2. Abstract lines 46-49 "Whereas Ov-grn-1 gene-edited parasites colonized the biliary tract and developed into adult flukes, less hepatobiliary tract disease manifested during chronic infection with ΔOv-grn-1 worms in comparison to hamsters infected with control parasites". This doesn't make sense to me. Do the authors mean instead "Gene-edited parasites colonized the biliary tract and developed into adult flukes, however less hepatobiliary tract disease manifested during chronic infection with ΔOv-grn-1 worms in comparison to hamsters infected with control parasites" ?

3. Introduction lines 91-94: the impact of nitrosamines in fermented foods on carcinogenesis needs a citation.

4. Introduction lines 91-94: It is unhelpful and somewhat lazy to repeat the mantra that "The mechanism(s) by which opisthorchiasis leads to CCA is likely multi-factorial..." after the amount of evidence accumulated in this area. Some factors are invariably more important than others and their impact can be quantified. In your previous study (Arunsan 2019 Elife) what proportion of the variance in mutagenesis was explained by knockdown of Ov-grn-1? How much of the variance can be explained by differential worm burdens? The authors should be attempting to quantify the relative importance of these different exposures in their experimental system.

5. Introduction lines 106-8: "Also, we recently confirmed the role of Ov-GRN-1 in driving proliferation of bile duct epithelial cells (cholangiocytes) by genetic manipulation of its expression in the liver fluke both by RNAi and by CRISPR/Cas9 gene editing (7)." The cited study (Arunsan 2019 Elife) appears similar in content to this manuscript. There needs to be additional explanation for the limitations in previous work which are overcome by the present study and what these experiments add to current understanding.

6. Results line 135: The abbreviation of "newly excysted juveniles (NEJs)" adds to a number of acronyms in the paper and is unlikely to be familiar to many readers. Consider replacing this acronym with "juveniles" throughout.

7. Results line 137: This is the first mention of Ov-tsp-2. What is the function of this gene and why is it being knockout out in addition to Ov-grn-1? This should be explained in the introduction.

8. Results line 137: The use of "control" throughout, rather than "SCR" (scramble) would improve readability of the manuscript

9. Figure 1A: This figure could certainly be improved for clarity. Consider revising the tubes and lighting bolts on the left hand side. There are no images of O. viverrini worms, which makes it seem like the hamsters are being gene edited rather than the parasites. The rationale for the two experiments are also unclear at this stage.

10. Figures 1B and C: There are a number of issues with both this figure and the analysis of these data. Presented as a "percentage of control abundance" is confusing as the SCR samples themselves vary. Presumably "100% of control" is the mean abundance in the SCR group? The following recommendations are: A) present the data on the original scale (number of transcripts), B) show each data point individually, rather than as a bar chart, C) remove the references to significance testing (stars and NS), D) replace the label "SCR" with "control", E) remove "NEJ" text on y axis.

11. Relating to this, the statistical tests performed to demonstrate gene knockout are unclear and likely inappropriate. Was the "one-way ANOVA with Holm-Sidak multiple comparison" performed on the original transcript abundance data, or after transformation into a relative proportion? I would strongly recommend against transforming the data prior to analysis. The ANOVA results (effect size) are also not shown. Given the small number of samples it would be better to use a statistical procedure other than ANOVA. Please calculate 95% confidence intervals within each experimental group by bootstrapping the transcript abundance data and check for overlap. The 95% confidence intervals can then be overlaid over the data points in Figures 1B and C, as recommended above.

12. Figures 2A and 2B, show all data points on the plots and remove text relating to significance testing (stars and NS). How are the values in Figures 2C and D different to Figures 1B and C? According to the text, these are all results from "experiment 1". Please show the raw data points rather than violin plots. Again I ask to authors to calculate 95% confidence intervals by bootstrapping values of untransformed transcript abundance.

13. Results lines 183-4: "183 Transcript levels of both genes expressed by the control SCR parasites were generally clustered

184 around 100% (Figure 2C, D)." Again, what is 100%? Is this the mean transcript abundance in the SCR group? In which case it's unsurprising that the relative values in this control group would be close to the mean. Instead provide the variance in transcript abundance on an untransformed scale.

14. Results line 187-9: "The outcome, where most flukes either not showing a change (~100%) or, by contrast, showing a near absence of transcription (~0%) was not normally distributed, which required a non-parametric statistical approach". This does not make sense. The parametric distribution for an overdispersed proportion is the beta-binomial; there is no need to use non-parametric methods. As mentioned before, the counts of transcript abundance should not be transformed into relative proportions prior to analysis. Until this analysis is re-performed, the changes in transcript abundance given in the text are difficult to interpret.

15. Results line 196-7: Is it a "lethal phenotype" if 1/3 of knockout worms survive?

16. Results line 223-22: "In lieu of counting the number of worms parasitizing each hamster, fecal egg count (as eggs per gram of feces, EPG) were determined at the time of necropsy. Numbers of eggs of O. viverrini positively correlate with number of worms within the hepatobiliary tract (12, 13)". The citations given here are both inappropriate as they relate to human studies (and one is in Clonorchis), rather than an experimental hamster system. The authors have the data to show the relationship between egg counts and worm burden from experiment 1 (from Figure 1: EPG values 1 and 2, and worm counts at necropsy). Please plot these data (egg counts vs worm burden) and calculate the relationship between them - note that the relationship may be non-linear due to density dependent fecundity (eggs per worm decrease at high worm burdens) and so the analysis may require a non-linear function (e.g. power law).

17. Results lines 232-238: "Hence, to mitigate the impact of variations in numbers of liver flukes on the analysis and interpretation of the role of gene knockout on hepatobiliary disease and malignancy, the histopathological assessments focused on the livers of hamsters with “moderate” EPG values, i.e. ΔOv-grn-1 EPG ranging from 1,000 to 20,000 (Figure 3A, B, dotted lines). This cutoff window excluded three SCR hamsters with values of 26,000 to 44,000 EPG and four ΔOv-tsp-2-infected hamsters with ≤238 1,000 EPG". This is completely inappropriate. The statistical analysis should correct for the number of parasites (either EPG or inferred worm burden given the relationship estimated above) in a multiple regression model. This would have the advantage of disentangling the effect of pathology due to worm burden versus knockdown of granulin. Excluding animals with arbitrary cutoffs of EPG reduces the power of the analysis, is unethical given the hamsters have been euthanised, and has no basis in experimental design.

18. Results 279-281: "Substitution patterns as determined by the CRISPR-sub tool (15) in reads in the knockout groups were not significantly different from the cognate alleles in the control SCR group worms (Figure S1)." This supplementary figure is confusing and does not substantiate the claim that read depth is the same across study groups. Please clarify the y-axis as a value cannot be both a rate and a percentage. The x-axis is very unclear - which region of the gene / genome is this? The authors should instead show read depth plots from a genome browser with variants marked, and report the number of mapped reads in the region of interest across samples.

19. Results 284-285: "The ΔOv-grn-1 pooled NEJs showed 3.26% indel levels (2,723 of 80,571 reads), significantly

285 more than the 0.035% level in the SCR NEJ group (18 of 51,402 reads) (P≤ 0.05)." This does not make sense. Indels are reported as a function of read depth per sample. Please demonstrate that indels have been called correctly and give the proportion of knockdown samples with indels in the gene regions of interest, rather than the proportion of reads.

20. Results line 298: 3.1% of what?

21: How are indels in knockout worms assessed if the adult flukes cannot be recovered from euthanised hamsters?

22. The differences in pathology reported in the subsequent results sections (fibrosis and collagen) could be confounded by different numbers of adult worms between the control group and knockdown; this should be controlled for in a regression model.

PLOS authors have the option to publish the peer review history of their article (what does this mean?). If published, this will include your full peer review and any attached files.

Reviewer #1: **Yes: **P'ng Loke

Reviewer #2: No

Reviewer #3: No

Figure Files:

Data Requirements:

Reproducibility:

References:

---

## [Editor Report · Decision Letter 1]

29 Aug 2022

Dear Brindley,

We are pleased to inform you that your manuscript 'Knockout of liver fluke granulin, Ov - grn - 1 , impedes malignant transformation during chronic infection with Opisthorchis viverrini' has been provisionally accepted for publication in PLOS Pathogens.

Best regards,

Kenneth D Vernick

Associate Editor

PLOS Pathogens

P'ng Loke

Section Editor

PLOS Pathogens

Kasturi Haldar

Editor-in-Chief

PLOS Pathogens

orcid.org/0000-0001-5065-158X

Michael Malim

Editor-in-Chief

PLOS Pathogens

orcid.org/0000-0002-7699-2064
---

## [Editor Report · Acceptance letter]

16 Sep 2022

Dear Brindley,

We are delighted to inform you that your manuscript, "Knockout of liver fluke granulin, Ov - grn - 1, impedes malignant transformation during chronic infection with Opisthorchis viverrini

," has been formally accepted for publication in PLOS Pathogens.

Best regards,

Kasturi Haldar

Editor-in-Chief

PLOS Pathogens

orcid.org/0000-0001-5065-158X

Michael Malim

Editor-in-Chief

PLOS Pathogens

orcid.org/0000-0002-7699-2064